# The ectodomains determine ligand function in vivo and selectivity of DLL1 and DLL4 toward NOTCH1 and NOTCH2 in vitro

Lena Tveriakhina[1], Karin Schuster-Gossler[1], Sanchez M Jarrett[2], Marie B Andrawes[2†], Meike Rohrbach[1], Stephen C Blacklow[2,3]*, Achim Gossler[1]*

[1]Institute for Molecular Biology, Medizinische Hochschule Hannover, Hannover, Germany; [2]Department of Biological Chemistry and Molecular Pharmacology, Harvard Medical School, Boston, Massachusetts; [3]Department of Cancer Biology, Dana Farber Cancer Institute, Boston, Massachusetts

**Abstract** DLL1 and DLL4 are Notch ligands with high structural similarity but context-dependent functional differences. Here, we analyze their functional divergence using cellular co-culture assays, biochemical studies, and in vivo experiments. DLL1 and DLL4 activate NOTCH1 and NOTCH2 differently in cell-based assays and this discriminating potential lies in the region between the N-terminus and EGF repeat three. Mice expressing chimeric ligands indicate that the ectodomains dictate ligand function during somitogenesis, and that during myogenesis even regions C-terminal to EGF3 are interchangeable. Substitution of NOTCH1-interface residues in the MNNL and DSL domains of DLL1 with the corresponding amino acids of DLL4, however, does not disrupt DLL1 function in vivo. Collectively, our data show that DLL4 preferentially activates NOTCH1 over NOTCH2, whereas DLL1 is equally effective in activating NOTCH1 and NOTCH2, establishing that the ectodomains dictate selective ligand function in vivo, and that features outside the known binding interface contribute to their differences.
DOI: https://doi.org/10.7554/eLife.40045.001

*For correspondence:
stephen_blacklow@hms.harvard.edu (SCB);
gossler.achim@mh-hannover.de (AG)

Present address: †Evidera, Waltham, United State

Competing interests: The authors declare that no competing interests exist.

## Introduction

The Notch signaling pathway mediates communication between neighboring cells in metazoans and thereby regulates a multitude of developmental processes in various tissues (*Artavanis-Tsakonas et al., 1995*; *Yoon and Gaiano, 2005*; *Bolós et al., 2007*; *Gridley, 2007*; *Radtke et al., 2010*; *Koch and Radtke, 2011*; reviewed in *Louvi and Artavanis-Tsakonas, 2012*; *Kopan, 2012*). This communication depends on the interaction of Notch receptors on the surface of the signal receiving cells with transmembrane ligands on the surface of adjacent cells. Ligand binding then leads to a sequence of proteolytic cleavages of the receptor releasing the Notch intracellular domain (NICD) from the membrane. NICD translocates into the nucleus where it enters into a complex with a CSL protein (CBF-1/RBPJ in mammals, Suppressor of Hairless in flies, and Lag-1 in worms) and a protein of the Mastermind family (*Petcherski and Kimble, 2000*; *Wu et al., 2000*; *Nam et al., 2003*; *Nam et al., 2006*; *Wilson and Kovall, 2006*; *Choi et al., 2012*) to regulate transcription of target genes (reviewed in *Bray, 2016*).

Mammals have four Notch receptors (N1-N4) and four activating ligands of the DSL (Delta, Serrate, LAG-2) family: DLL1 and DLL4, orthologs of *Drosophila* Delta, and JAG1 and JAG2, orthologs of *Drosophila* Serrate. DLL1 and DLL4 are similar in domain structure, size and sequence (*Shutter et al., 2000*). Both proteins contain an N-terminal MNNL (also referred to as C2) domain

**eLife digest** A small number of signaling systems control how an animal develops from a single cell into a complex organism made up of many different cell types. Signals pass back and forth between cells, switching genes on and off to direct the development of tissues and organs. One of these signaling systems, called Notch, is so ancient that it appears in nearly all multicellular organisms.

A cell sends a Notch signal using proteins called Delta or Jagged ligands that span membrane of the cell, so that part of the protein sits inside the cell and part remains outside. To change the behavior of another cell, the ligands bind to proteins called Notch receptors that span the membrane of the receiving cell.

Mammals have two types of Delta ligand, two types of Jagged ligand and four types of Notch receptor. Cells in different tissues display different combinations of these eight proteins. Two Delta ligands called DLL1 and DLL4 often appear together in developing organisms. Some tissues need both and some only the one or the other. In some cases one ligand can compensate if the other is missing, but in others not. It was not clear why this is, or which parts of the proteins are responsible.

Tveriakhina et al. used mouse cells to investigate how DLL1 and DLL4 interact with two Notch receptors, called NOTCH1 and NOTCH2. The results of these experiments show that while DLL1 can bind and activate both Notch receptors equally, DLL4 prefers to partner with NOTCH1. To find out which parts of the ligands are responsible for this selectivity, Tveriakhina et al. created hybrid ligands that contained a mixture of regions from DLL1 and DLL4. These suggest that the different binding preferences depend on parts of the ligands that sit outside cells and that lie outside the known sites of binding contact with the Notch receptors.

Further experiments studied mice that had been engineered to produce hybrid ligands as replacements for DLL1. A hybrid ligand consisting of the part of DLL1 that sits outside cells and the part of DLL4 found inside cells generated Notch signals in the tissue that depended on the activity of DLL1. However, a hybrid consisting of the part of DLL4 that sits outside cells and the part of DLL1 found inside cells did not, showing that in developing mice the parts that sit outside the cells contribute to the different functions of DLL1 and DLL4.

Overall, the results presented by Tveriakhina et al. show that interactions between specific ligands and receptors play important roles in how mammals develop. Further efforts to understand which parts of the ligands affect selectivity could ultimately allow researchers to develop ways to modify how ligands and receptors interact. Such "molecular engineering" strategies could enable cell responses to be precisely controlled by pairing designer ligand-receptor pairs to develop cell-based therapies.

DOI: https://doi.org/10.7554/eLife.40045.002

(*Chillakuri et al., 2013*; *Suckling et al., 2017*), followed by a DSL domain and eight EGF-like repeats in their extracellular portion, and a less well conserved intracellular domain. The MNNL and DSL domains, required for high-affinity binding of Delta-like ligands to Notch receptors (*Rebay et al., 1991*; *Cordle et al., 2008*), contact EGF repeats 12 and 11 of Notch, respectively (*Luca et al., 2015*). Contributions from adjacent EGF-like repeats, however, are required for signal transduction by Delta-like ligands (*Andrawes et al., 2013*) as well as for optimal interaction with Serrate (*Yamamoto et al., 2012*) and Jagged (JAG)-family ligands (*Luca et al., 2017*). Although the biological activities of DLL1 and DLL4 are partially overlapping, the two proteins are not equivalent in vitro or in vivo. In cell culture studies, DLL4 is more effective than DLL1 in activating N1 signaling during T cell development (*Besseyrias et al., 2007*), consistent with its ten-fold higher binding affinity in binding studies using purified fragments of N1, DLL1, and DLL4 (*Andrawes et al., 2013*). In vivo, studies of adult intestinal epithelium in mice have shown that DLL1 and DLL4 are co-expressed in crypts and act redundantly to maintain the intestinal stem cell pool (*Pellegrinet et al., 2011*). In contrast, however, mouse DLL1 cannot fully replace DLL4 in its ability to trigger T lineage commitment (*Besseyrias et al., 2007*; *Mohtashami et al., 2010*). Conversely, endogenous DLL4 does not substitute for DLL1 in its ability to promote development of the arterial vascular epithelium (*Sörensen et al., 2009*), nor does it compensate for the function of DLL1 in the paraxial mesoderm,

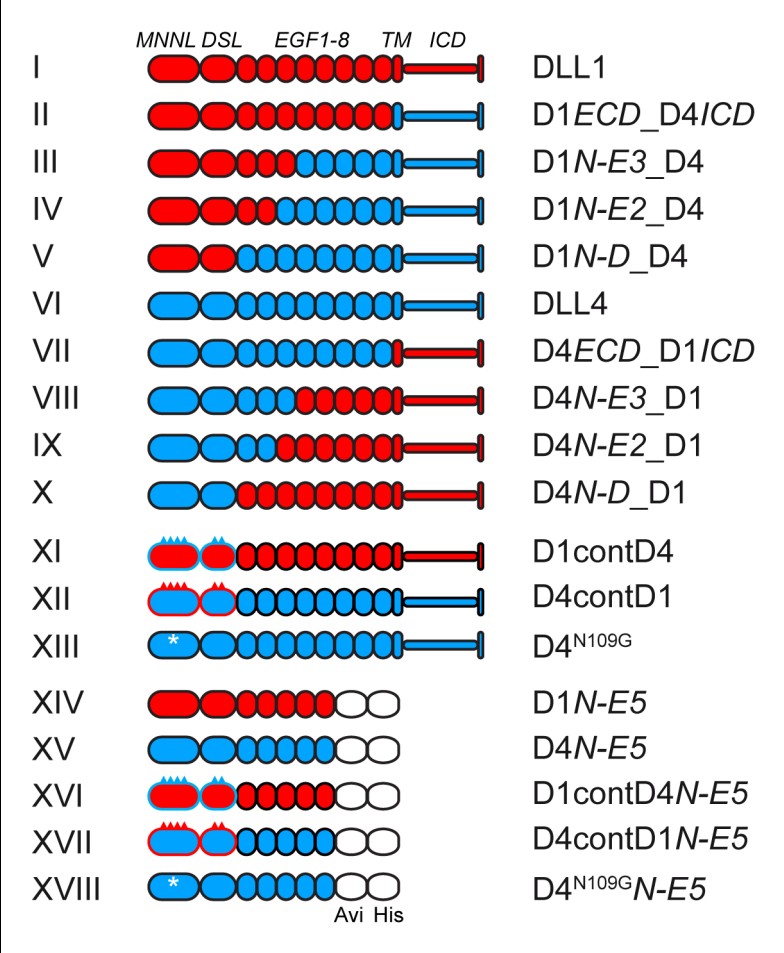

**Figure 1.** Schematic representation of DLL1 and DLL4 and variant proteins. I-X, full-length and chimeric ligands generated by domain swaps. XI and XII, ligands with exchanges of the known NOTCH1 contact amino acids in the MNNL and DSL domains. XIII, DLL4 variant with an N109G mutation that eliminates the N-glycosylation site in DLL4. XIV-XVIII, soluble proteins encoding the N-terminal region up to and including EGF5 carrying a C-terminal Avi-His-tag for protein purification. I-XIII were tested in cell-based Notch activation assays, II, III, VII and XI in transgenic mice, XIV-XVIII used for measurements of binding affinities to N1. Proteins analyzed in cell-based assays were C-terminally Flag-tagged, proteins analyzed in mice were untagged. Break points and surrounding amino acid sequences and point substitutions are illustrated in *Figure 1—figure supplement 1*. Red domains/spikes: DLL1; blue domains/spikes: DLL4; white asterisks: N109G mutation. *ECD*, extracellular domain; *N*, N-terminus; *D*, DSL domain; *E*, EGF repeat, *TM*, transmembrane domain; *ICD*, intracellular domain; D, DLL; cont, N1 contact amino acids.

DOI: https://doi.org/10.7554/eLife.40045.003

The following figure supplements are available for figure 1:

**Figure supplement 1.** Amino acid exchanges of DLL variant proteins.
DOI: https://doi.org/10.7554/eLife.40045.004

**Figure supplement 2.** Analysis of ligand receptor binding.
DOI: https://doi.org/10.7554/eLife.40045.005

**Figure supplement 3.** N109 is highly conserved and N-glycosylated in DLL4.
DOI: https://doi.org/10.7554/eLife.40045.006

a tissue where these ligands are normally not co-expressed: mice in which DLL1 was replaced by DLL4 had severe somite patterning defects and showed premature myogenic differentiation leading to reduced skeletal muscles. However, the function of DLL1 during early retina development was rescued by DLL4 in these mice (*Preuße et al., 2015*).

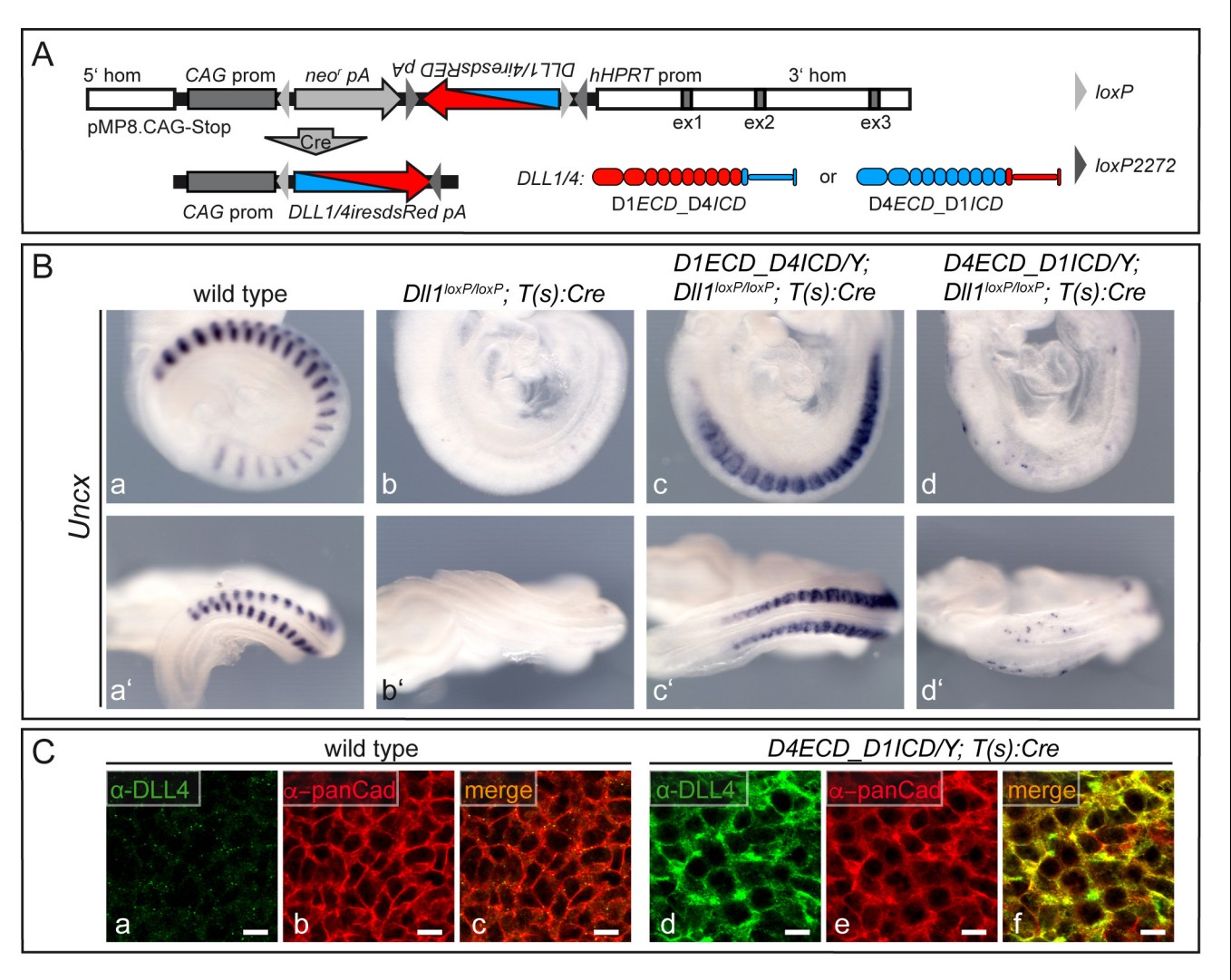

**Figure 2.** The extracellular domains of DLL1 and DLL4 determine ligand behavior during somitogenesis. (**A**) Scheme of the targeting vector pMP8. CAG-Stop used to introduce inducible chimeric ligands into the *Hprt* locus, and of Cre-mediated activation of transgene (D1*ECD*_D4*ICD* or D4*ECD*_D1*ICD*) expression driven by the CAG promotor (*CAG* prom). 5' hom and 3' hom, *Hprt* 5' and 3' homology regions; ex1-3 (grey boxes), *Hprt* exons; *neo*[r], neomycin phosphotransferase; *pA*, polyadenylation signal; *hHPRT* prom, human *Hprt* promoter; *DLL1/4iresdsRED*, chimeric ORF–linked to dsRed tag by an internal ribosomal entry site (IRES). (**B**) *Uncx* expression in E9.5 wild type embryos (a, a'; n = 28), embryos lacking DLL1 in the mesoderm (b, b'; n = 12) and male embryos lacking DLL1 in the mesoderm that express either D1*ECD*_D4*ICD* (c, c'; n = 9) or D4*ECD*_D1*ICD* (d, d'; n = 8) showing that the extracellular domain of DLL1 but not of DLL4 can restore *Uncx* expression. (**C**) Whole mount immunofluorescent staining of wild type (a–c) and *D4ECD_D1ICD/Y;T(s):Cre* (d–f) PSMs using antibodies recognizing the extracellular domain of DLL4 showing co-localization of the exogenous chimeric ligand with pan-Cadherin (panCad) at the cell surface. Additional intracellular staining most likely reflects the presence of the ligand in the ER and trans Golgi as observed previously for DLL1 in cultured cells (*Geffers et al., 2007*; *Müller et al., 2014*) and for endogenous DLL1 and transgenic DLL4 in the PSM (*Preuße et al., 2015*). n = 3 for wild type, n = 4 for *D4ECD_D1ICD/Y;T(s):Cre*; Scale bar = 10 μm.
DOI: https://doi.org/10.7554/eLife.40045.007

Collectively these studies indicate that the functionality of DLL1 and DLL4 strongly depends on context, but it remains unclear which portions of these similar DSL proteins account for their functional non-equivalence. A recent study in cell culture observed that DLL1 and DLL4 stimulate NOTCH1 receptors to produce responses with different dynamics, attributing differences between pulsatile signaling of DLL1 and sustained signaling by DLL4 to the intracellular, rather than the extracellular, regions of the proteins (*Nandagopal et al., 2018*). Here, we investigate the influence of the extracellular and intracellular regions of DLL1 and DLL4 chimeric proteins on ligand function in cell culture assays, and for selected chimeras, in biochemical binding assays and in vivo in mice. We

observe that in vivo differences of DLL1 and DLL4 function during somite patterning and myogenesis are encoded by the ligands ectodomains, that DLL1 and DLL4 are able to discriminate between NOTCH1 and NOTCH2 in vitro, and that ligand residues outside of the known binding interface are important contributing factors for ligand function in vivo.

## Results

### The extracellular domain dominates ligand function during somitogenesis

Previous in vivo analyses indicated that DLL4 cannot substitute for DLL1 function during embryonic development (*Preuße et al., 2015*). To test whether the inability of DLL4 to rescue the loss of DLL1 in the paraxial mesoderm in vivo resides in its extra- or intracellular domain we generated single copy transgenic mice allowing for the conditional expression of chimeric DLL molecules consisting of the extracellular domain of one ligand and transmembrane and intracellular domain of the other (D1*ECD*_D4*ICD* and D4*ECD*_D1*ICD*, II and VII in *Figure 1*). Transgenes were introduced into the *Hprt* locus of *Hprt*-deficient E14TG2a ES cells by homologous recombination using the strategy already employed for the initial analysis of full length DLL1 and DLL4 during somitogenesis (*Preuße et al., 2015*). Briefly, cDNAs encoding chimeric ligands were cloned into the targeting vector pMP8 in reverse orientation downstream of neomycin phosphotransferase (*neo*^r) driven by the CAG promoter. Cre-mediated recombination of two *loxP* sites and two mutant *loxP2272* (*loxM*) sites removes the *neo*^r cassette and flips the gene of interest and results in its expression from the CAG promoter (*Figure 2A*).

To test whether the extracellular or intracellular domain determines the inability of DLL4 to rescue the loss of DLL1 in mesodermal tissues of early embryos, we induced expression of either chimeric ligand and simultaneously removed endogenous DLL1 using a floxed *Dll1* allele and a Cre transgene expressed in the primitive streak driven by a promoter derived from *brachyury* (*T(s):Cre*) (*Feller et al., 2008*). Because the *Hprt* locus is located on the X-chromosome, we used hemizygous male embryos for the analysis. As previously described, inactivation of *Dll1* in the mesoderm resulted in loss of *Uncx* (formerly called *Uncx4.1*) expression in caudal somite compartments (n = 12; *Figure 2Bb,b'*) indicating severe somite patterning defects compared to wild type embryos (n = 28; *Figure 2Ba,a'*). Expression of D1*ECD*_D4*ICD* in *Dll1*-deficient embryos (n = 9) restored robust expression of *Uncx* similar to full length DLL1 (*Preuße et al., 2015*). *Uncx* expression expanded into cranial somite compartments (*Figure 2Bc,c'*) reminiscent of ectopic Notch activity (*Feller et al., 2008*), probably reflecting non-restricted D1*ECD*_D4*ICD* expression throughout the PSM and somites. In contrast, expression of D4*ECD*_D1*ICD* barely restored *Uncx* expression in the majority (n = 8/12) of *Dll1*-deficient embryos (*Figure 2Bd,d'*), a phenotype similar to that seen with full-length DLL4 (*Preuße et al., 2015*), even though the chimeric ligand was expressed and detected on the cell surface of PSM cells (*Figure 2Cd-f*). As observed previously for full-length DLL4 (*Preuße et al., 2015*) some embryos (n = 4) displayed essentially normal *Uncx* expression (not shown), which might result from some perdurance of DLL1 activity or delayed or inefficient excision of endogenous *Dll1*. Overall, this analysis strongly suggests that the functional difference between DLL1 and DLL4 observed in vivo during somitogenesis resides in the extracellular domains.

### Regions outside the known receptor binding domain are essential for full DLL1 function in vivo

The N-terminal MNNL and DSL domains and adjacent EGF repeats 1–3 constitute the major interface for interaction between DSL ligands and Notch receptors, and are essential for (full) activation of Notch signaling (*Cordle et al., 2008*; *Andrawes et al., 2013*; *Luca et al., 2015*; *Schuster-Gossler et al., 2016*; *Luca et al., 2017*). To analyze whether this region accounts for the observed differences between DLL1 and DLL4 in vivo we generated a chimeric ligand that contained the N-terminal region up to and including EGF3 of DLL1 fused to EGF4 and the remaining C-terminal portion of DLL4 (D1*N-E3*_D4, III in *Figure 1*; the amino acid sequence around the fusion is shown in *Figure 1—figure supplement 1Ab*). We then tested whether this chimeric ligand is sufficient for normal DLL1 function during development. We generated mice (*Dll1*^D1N-E3_D4ki^) expressing D1*N-E3*_D4 instead of DLL1 using the "mini-gene" knock-in strategy (*Figure 3A*) that disrupts endogenous *Dll1*,

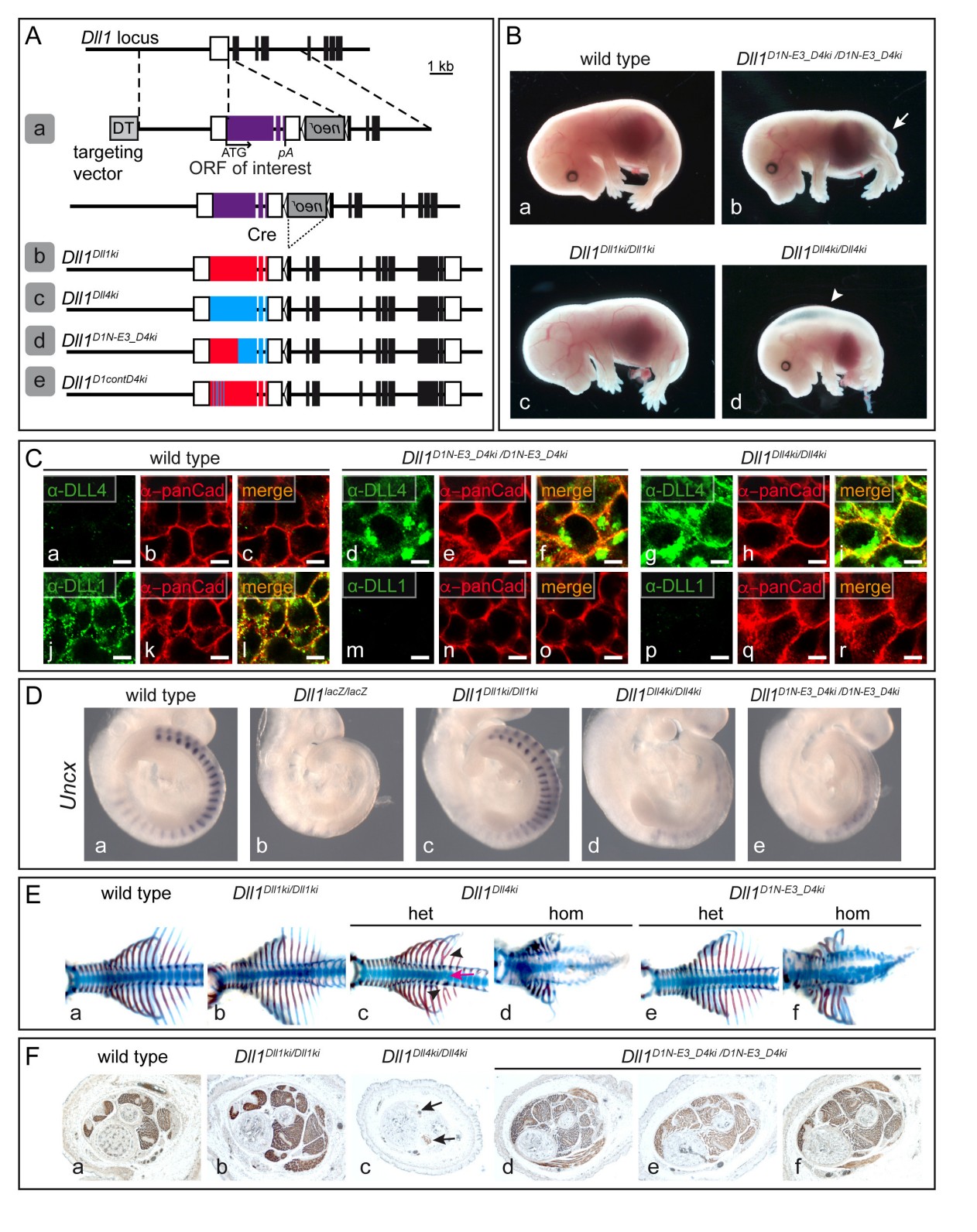

**Figure 3.** D1N-E3_D4 is not able to compensate for DLL1 function during somitogenesis. (**A**) "Mini-gene" targeting strategy to express DLL1 or DLL4 variants from the *Dll1* locus (a) and alleles generated in this study (**d and e**). The *Dll1^Dll1ki^* (**b**) and *Dll1^Dll4ki^* (**c**) control alleles were described previously (**Preuße et al., 2015**; **Schuster-Gossler et al., 2016**). *Dll1^D1N-E3_D4ki^* (**d**) encodes a fusion protein between the N-terminal part of DLL1 including EGF3 fused to EGF4 and the remaining C-terminal portion of DLL4 (III in **Figure 1** and **Figure 1—figure supplement 1Ab**). *Dll1^D1contD4ki^* (**e**) encodes a DLL1

*Figure 3 continued on next page*

Figure 3 continued

variant whose predicted amino acids of the MNNL and DSL domains that contact N1 are replaced by the corresponding amino acids of DLL4 (XI in *Figure 1*, *Figure 5C*, and *Figure 1—figure supplement 1B*). All alleles have an identical structure and intron 9 and 10 sequences of *Dll1*. (B) External phenotypes of wild type (a; n = 19), homozygous *Dll1^{D1N-E3_D4ki}* (b; n = 11), *Dll1^{Dll1ki}* (c; n = 3) and *Dll1^{Dll4ki}* (d; n = 3) control E15.5 fetuses. Arrow in (b) points to the short tail. Arrowhead in (c) points to edemas present in homozygous *Dll1^{Dll4ki}* fetuses. (C) Indirect immunofluorescence staining of wild type (a–c, j–l), homozygous *Dll1^{D1N-E3_D4ki}* (d–f, m–o), and homozygous *Dll1^{Dll4ki}* (g–i, p–r) E9.5 PSMs using antibodies recognizing the extracellular domain of DLL4 (a, d, g) and DLL1 (j, m, p) and pan-Cadherin (panCad; b, e, h, k, n, q) showing expression of D1N-E3_D4 and co-localization with the cell surface marker pan-Cadherin. Staining of D1N-E3_D4 appears weaker than DLL4 most likely because much of the epitope recognized by the polyclonal anti-DLL4 antibody is missing in this chimeric protein. n ≥ 3; Scale bar = 5 μm. (D) WISH of E9.5 embryos showing that D1N-E3_D4 does not restore normal *Uncx* expression (e; n = 10) resembling the *Dll1^{Dll4ki}* phenotype (d; n = 7). (E) Skeletal preparations of wild type (a; n = 11), homozygous *Dll1^{Dll1ki}* (b; n = 6), heterozygous (c; n = 14/16) and homozygous (d; n = 3) *Dll1^{Dll4ki}*, and heterozygous (e; n = 14) and homozygous (f; n = 10) *Dll1^{D1N-E3_D4ki}* E15.5 fetuses. Arrow and arrowheads in (c) point to axial skeleton defects that were not detected in *Dll1^{D1N-E3_D4ki}* heterozygotes (e). (F) Cross-sections of hind limbs of wild type (a), homozygous *Dll1^{Dll1ki}* (b), homozygous *Dll1^{Dll4ki}* (c), and homozygous (d-f; n = 3) *Dll1^{D1N-E3_D4ki}* E18.5 fetuses stained for expression of Myosin Heavy Chain (MHC) indicating that D1N-E3_D4 rescues the skeletal muscle phenotype in contrast to DLL4. Arrows in (c) point to skeletal muscle remnants.

DOI: https://doi.org/10.7554/eLife.40045.008

successfully employed previously to express either a *Dll4* or *Dll1* (control) mini-gene (*Schuster-Gossler et al., 2007*; *Preuße et al., 2015*; *Schuster-Gossler et al., 2016*). Heterozygous mice obtained from two independent targeting events carrying the *Dll1^{D1N-E3_D4ki}* allele were viable and showed no apparent phenotype. Homozygous *Dll1^{D1N-E3_D4ki}* mice were stillborn (n = 3 and 4, respectively), indicating that D1N-E3_D4 cannot fully replace DLL1 during development although it is present on the cell surface of PSM cells (*Figure 3Cd-f*). At E15.5 homozygous *Dll1^{D1N-E3_D4ki}* fetuses showed a stumpy tail (n = 5 and 6, respectively; arrow in *Figure 3Bb*) similar to *Dll1^{Dll4ki}* mutants (*Figure 3Bd*); however, they lacked the edema observed in *Dll1^{Dll4ki}* homozygotes (arrowhead in *Figure 3Bd*). D1N-E3_D4 was also not able to restore normal *Uncx* expression (*Figure 3De*). Axial skeletons of homozygous *Dll1^{D1N-E3_D4ki}* fetuses were severely disorganized (n = 10; *Figure 3Ef*), a phenotype consistent with abnormal *Uncx* expression and similar to *Dll1^{Dll4ki/Dll4ki}* homozygote axial skeletons (*Figure 3Ed*), although the rib cage appeared less compressed. In contrast to *Dll1^{Dll4ki}* heterozygotes (*Figure 3Ec*; *Preuße et al., 2015*), which often displayed axial skeleton defects (n = 14/16) such as hemivertebrae (arrow in *Figure 3Ec*) and fused ribs (arrowheads in *Figure 3Ec*) heterozygous *Dll1^{D1N-E3_D4ki}* fetuses showed no defects of the axial skeleton (n = 0/14; *Figure 3Ee*) indicating that D1N-E3_D4 lacks the dominant interfering activity of DLL4.

Deletion of DLL1 during myogenesis leads to premature differentiation of myogenic progenitor cells resulting in severe skeletal muscle hypotrophy at fetal stages (*Schuster-Gossler et al., 2007*). This phenotype cannot be suppressed by DLL4 expression (*Figure 3Fc*; *Preuße et al., 2015*). In contrast, skeletal muscles of *Dll1^{D1N-E3_D4ki/D1N-E3_D4ki}* homozygotes (*Figure 3Fd-f*; n = 3) were virtually indistinguishable from *Dll1^{Dll1ki/Dll1ki}* (*Figure 3Fb*) and wild type fetuses (*Figure 3Fa*). These in vivo analyses indicate that, unlike the D1ECD_D4ICD chimera, D1N-E3_D4 is not a fully functional DLL1 ligand during somite patterning. However, D1N-E3_D4 remains functional during myogenesis and restricts muscle progenitor differentiation despite the presence of the DLL4 ICD, consistent with the conclusion that in vivo the functional difference between DLL1 and DLL4 is encoded in the ECDs.

## DLL1 and DLL4 exhibit differential receptor selectivity in vitro

In cell-based trans-activation assays using HeLa cells stably expressing murine N1 (HeLaN1) co-cultured with CHO cells expressing mouse DLL1 (mDLL1) or DLL4 from the same locus both ligands activated a transiently expressed Notch reporter similarly (*Preuße et al., 2015*). However, a purified fragment of the extracellular domain of human DLL4 (N-terminus up to and including EGF5: hD4N-E5) bound to hN1 with an approximately ten-fold higher affinity than the corresponding hDLL1 fragment (*Andrawes et al., 2013*). Like hD4N-E5, mD4N-E5 has a higher affinity for hN1 ($K_D = 0.43 \pm 0.046$ μM; *Figure 1—figure supplement 2Aa*) than the corresponding mDLL1 fragment ($K_D = 1.56 \pm 0.207$ μM; *Figure 1—figure supplement 2Ab*), as judged by biolayer interferometry measurements. To find a potential explanation for the discrepancy between binding affinities and Notch activation in HeLaN1 cells we analyzed these cells for expression of other Notch receptors and found that in addition to exogenous mouse *Notch1* HeLaN1 cells express endogenous *NOTCH2*

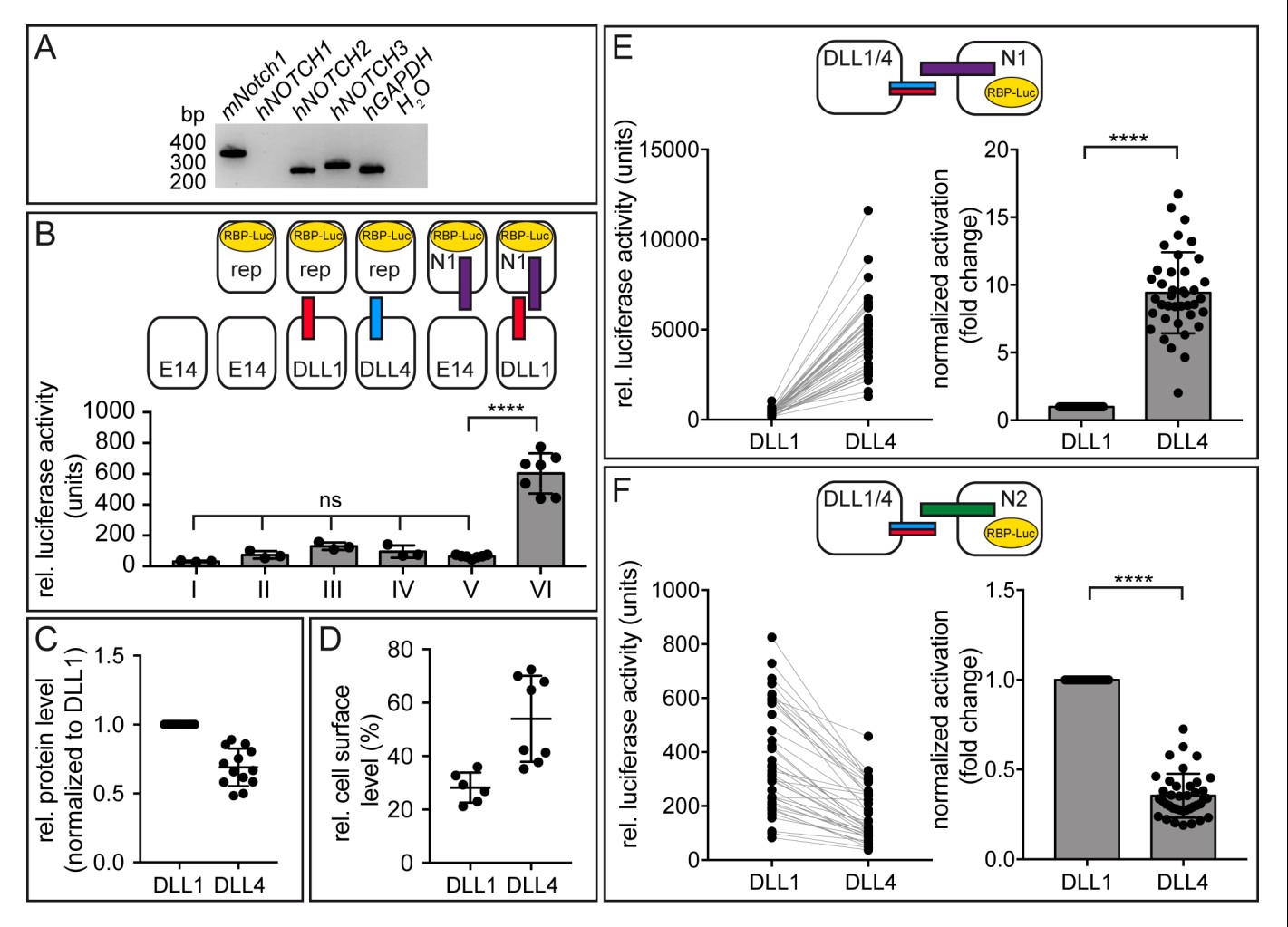

**Figure 4.** DLL1 and DLL4 differentially activate NOTCH1 and NOTCH2 in cell-based co-culture assays. (**A**) RT-PCR analysis using RNA of HeLaN1 cells shows the expression of endogenous human *NOTCH2* and *NOTCH3* in addition to the exogenous murine *Notch1*. (**B**) ES cell-based trans-activation assays demonstrate that E14TG2a ES cells express negligible amounts of endogenous Notch receptors and ligands. Co-cultivation of ES cells with stable expression of either DLL1 (III) or DLL4 (IV) from the *Hprt*-locus with ES-cells carrying only the RBP-Luc reporter in the *Hprt*-locus (E14rep) showed luciferase activity at levels indistinguishable from lysates of only E14 cells (I) and co-cultures of wild type E14 and reporter carrying ES cells (II). Similarly, co-culture of ES cells carrying N1 and the RBP-Luc reporter (N1rep) with E14 cells (V) did not show reporter activation significantly above background levels, whereas co-culture of DLL1 expressing cells with N1rep ES cells showed a 6–10-fold increase in luciferase activity (VI). n ≥ 3 co-cultures with 2–4 replicate measurements per n (***Figure 4—source data 1***). Mean ± SD, ns = p ≥ 0.05, ****=p < 0.0001, one-way ANOVA followed by Tukey's multiple comparison test. (**C**) Protein expression analysis indicating similar expression levels of DLL1 and DLL4 in the ES cell clones used. Each DLL4 value represents a technical replicate, which was referenced to its paired DLL1 value, which was arbitrarily set to one for each measurement. The non-normalized values (DLL/β-Tub ratios) are depicted in a graph in ***Figure 4—figure supplement 1A*** (***Figure 4—source data 2***). (**D**) Cell-surface biotinylation demonstrating that a slightly higher fraction of DLL4 is present at the cell surface compared to DLL1 (n ≥ 6; ***Figure 4—source data 3***). (**E**) DLL4 activates N1 about 10-fold more strongly than DLL1 in co-culture assays. Left graph shows non-normalized N1 activation. Lines connect values measured in the same assay. Right graph shows values normalized to DLL1 activation, and corrected for protein expression and cell surface presentation. (**F**) DLL4 activates N2 about half as strongly as does DLL1. Left graph shows non-normalized N2 activation. Lines connect values measured in the same assay. Right graph shows values normalized to DLL1 activation, and corrected for protein expression and cell surface presentation. Each dot represents a technical replicate. Raw data are shown in ***Figure 4—source data 4*** and ***Figure 4—source data 5***. Co-cultures (n = 39) with two replicate measurements per n. Mean ± SD, ns = p ≥ 0.05, ****=p < 0.0001, Student's paired t-test.

DOI: https://doi.org/10.7554/eLife.40045.009

The following source data and figure supplements are available for figure 4:

**Source data 1.** Raw data used to generate the graph in ***Figure 4B***.
DOI: https://doi.org/10.7554/eLife.40045.013

**Source data 2.** Data used to generate the graphs in ***Figure 4C*** and ***Figure 4—figure supplement 1A***.

*Figure 4 continued on next page*

*Figure 4 continued*

DOI: https://doi.org/10.7554/eLife.40045.014

**Source data 3.** Raw data used to generate the graph in *Figure 4D*.

DOI: https://doi.org/10.7554/eLife.40045.015

**Source data 4.** Numerical values used to generate the graphs in *Figure 4E*.

DOI: https://doi.org/10.7554/eLife.40045.016

**Source data 5.** Numerical values used to generate the graphs in *Figure 4F*.

DOI: https://doi.org/10.7554/eLife.40045.017

**Figure supplement 1.** Consistent N1 and N2 activation by different cell clones expressing DLL4.

DOI: https://doi.org/10.7554/eLife.40045.010

**Figure supplement 1—source Data 1.** Numerical values used to generate the graph in *Figure 4—figure supplement 1B*.

DOI: https://doi.org/10.7554/eLife.40045.011

**Figure supplement 1—source Data 2.** Numerical values used to generate the graphs in *Figure 4—figure supplement 1C,D*.

DOI: https://doi.org/10.7554/eLife.40045.012

and *NOTCH3* (*Figure 4A*), which might have masked underlying differences in the intrinsic N1 response to the DLL1 and DLL4 ligands.

To detect potential differences in ligand activity towards N1 or N2, the two Notch receptors present during somitogenesis, and to reduce variability due to transient reporter expression we stably integrated a Notch luciferase reporter in the *Hprt* locus (E14rep) of mouse E14TG2a ES (E14) cells, and generated stable cell lines expressing either *Notch1* (N1rep) or *Notch2* (N2rep) in these cells (*Schuster-Gossler et al., 2016*). When co-cultured with E14 cells or DLL1 or DLL4 expressing cells, E14rep cells show luciferase activity similar to wild type E14 levels (compare I to II, III, and IV in *Figure 4B*; numerical values in *Figure 4—source data 1*), indicating that negligible amounts of functional endogenous NOTCH receptors are present in E14 cells. Likewise, N1rep cells show essentially no activation above background when co-cultured with wild type E14 ES cells (compare V to I in *Figure 4B*; numerical values in *Figure 4—source data 1*), indicating insignificant amounts of functional endogenous Notch ligands in these cells. ES cells expressing exogenous DLL1 activate the luciferase reporter approximately ten-fold above the basal signal in E14 ES cells when co-cultured with N1rep cells (compare VI to V in *Figure 4B*; numerical values in *Figure 4—source data 1*) indicating that our co-culture system reliably measures specific Notch signaling activity.

To create ligand presenting cells for a comparison between mDLL1 and mDLL4, we generated ES cells expressing either mDLL1 or mDLL4 from single copy integrations into the *Hprt* locus. Co-cultures (n = 39) of cells expressing DLL1 or DLL4 with N1rep ES cells consistently revealed higher activation of N1 by DLL4 than by DLL1 (mean 12.454 ± 3.961 SD fold of non-normalized luciferase activity, 9.42 ± 2.997 SD fold, when normalized to DLL1 activation and corrected for protein expression and cell surface levels (*Figure 4C–E*; numerical values *Figure 4—source data 2*, *Figure 4—source data 3*, *Figure 4—source data 4*). In contrast, DLL4 activated N2 significantly less efficiently than did DLL1 (n = 39; mean 0.468 ± 0.161 SD fold of non-normalized luciferase activity, 0.35 ± 0.12 SD fold, when normalized to DLL1 activation and corrected for protein expression and cell surface levels (*Figure 4F*; numerical values in *Figure 4—source data 5*). To confirm that the observed differences between DLL1 and DLL4 in activating N1 and N2 were not a secondary consequence of clonal selection (however unlikely), we also analyzed additional DLL1 (n = 3) and DLL4 (n = 9) expressing ES cell clones for protein expression and N1 or N2 activation. Despite some variability of protein expression (*Figure 4—figure supplement 1B* and *Figure 4—figure supplement 1—source data 1*) and Notch activation levels between individual clones and co-cultures, all DLL4 clones consistently activated N1 significantly better than all DLL1 clones, and all DLL4 clones stimulated N2 significantly less efficiently than DLL1 (*Figure 4—figure supplement 1C,D*; numerical values in *Figure 4—figure supplement 1—Source Data 2*), indicating that both ligands differ significantly in their ability to activate different Notch receptors in our cell-based assay. Consistent with the differences in N2 stimulation by mDLL1- and mDLL4-expressing cells, the highly homologous human hD1*N-E5* exhibits a higher affinity ($K_D$ = 0.36 ± 0.11 μM; *Figure 1—figure supplement 2Ba*) for human NOTCH2 than D4*N-E5* ($K_D$ = 1.28 ± 0.2 μM; *Figure 1—figure supplement 2Bb*).

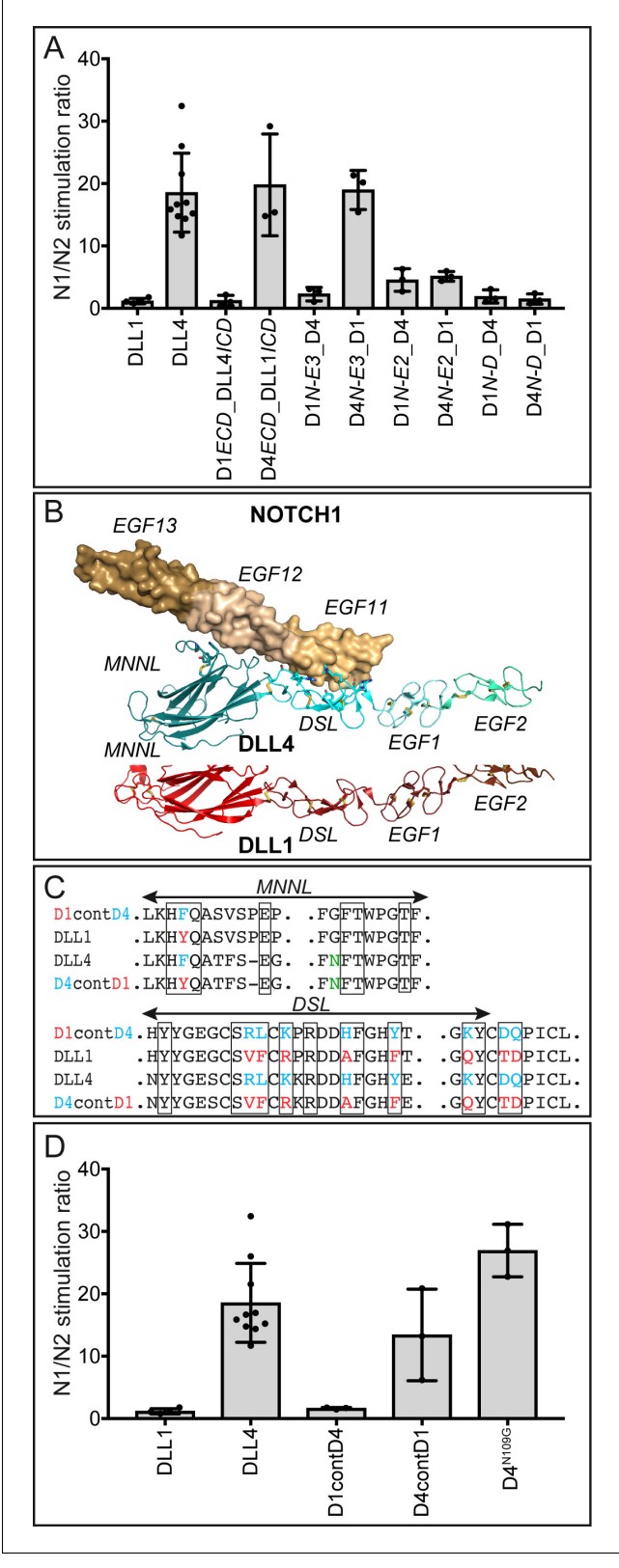

**Figure 5.** Contributions of the MNNL-EGF3 portion and contact amino acids to ligand selectivity towards N1 and N2. (**A**) N1/N2 activation ratios by DLL1 and DLL4 chimeric proteins show that receptor selectivity of DLL1 and DLL4 is encoded by the extracellular domain and that EGF3 contributes to N1/N2 selectivity. DLL4, D4*ECD*_D1*ICD*, and D4*N-E3*_D1 show N1/N2 induction ratios of ~20. DLL1, D1*ECD*_D4*ICD*, and D1*N-E3*_D4

*Figure 5 continued on next page*

*Figure 5 continued*

exhibit induction ratios of 1–3. Chimeric pairs with domain exchanges between EGF2 and EGF3 or between DSL domain and EGF1 show equivalent stimulation ratios. Each dot represents the mean of N1 (relative luciferase units; *Figure 5—source data 1*)/N2 (relative luciferase units; *Figure 5—source data 2*) of n ≥ 3 measurements per clone of a given ligand construct. Bars represent the Mean ± SD of n ≥ 3 clones per construct (*Figure 5—source data 3*). (B) Structure-based superposition of DLL1 and DLL4 (PDB ID codes 4XBM and 4XLW, respectively; (*Kershaw et al., 2015*; *Luca et al., 2015*). Top panel: NOTCH1 is rendered as a molecular surface (wheat), and DLL4 is rendered in ribbon representation (cyan). N1 contact residues on DLL4 were rendered as sticks, and were used to predict N1 contact amino acids of the MNNL and DSL domains of DLL1 (red). Domains are labeled above and below the structures, respectively, and individual domains are identified by different degrees of color shading/intensity. (C) Parts of the MNNL and DSL sequences showing the contact amino acids (boxed), the divergent amino acids of DLL1 (red) and DLL4 (blue), and the sequence of ligands with amino acid exchanges (complete sequences of the changed MNNL and DSL domains are shown in *Figure 1—figure supplement 1B*). The N-glycosylation site at residue N109 of DLL4 is indicated in green. (D) N1/N2 activation ratios of ligands with exchanged N1 contact amino acids. D1contD4 does not show changes in receptor selectivity compared to DLL1. Replacing the contact residues of DLL4 with those of DLL1 only reduces N1/N2 activation ratio to ~13. Elimination of the N-glycosylation site of DLL4 with the N109G mutation (the corresponding amino acid of DLL1) does not change DLL4 receptor selectivity. Each dot represents the mean of N1 (relative luciferase units; *Figure 5—source data 1*)/N2 (relative luciferase units; *Figure 5—source data 2*) of n ≥ 3 measurements per clone of a given ligand construct. Bars represent the Mean ± SD of n ≥ 3 clones per construct (*Figure 5—source data 3*).

DOI: https://doi.org/10.7554/eLife.40045.018

The following source data and figure supplement are available for figure 5:

**Source data 1.** Raw data (RLUs) of luciferase activity in co-cultures with N1rep cells used to generate the graph in *Figure 5—figure supplement 1A*.

DOI: https://doi.org/10.7554/eLife.40045.020

**Source data 2.** Raw data (RLUs) of luciferase activity in co-cultures with N1rep cells used to generate the graph in *Figure 5—figure supplement 1B*.

DOI: https://doi.org/10.7554/eLife.40045.021

**Source data 3.** N1/N2 activation ratios.

DOI: https://doi.org/10.7554/eLife.40045.022

**Figure supplement 1.** N1 and N2 activation by different ligand proteins.

DOI: https://doi.org/10.7554/eLife.40045.019

## The region encompassing the MNNL up to and including EGF3 encodes the differential receptor selectivity of DLL1 and DLL4

In an attempt to identify the domains of DLL1 and DLL4 that contribute to differences in activating N1 and N2, we carried out a series of domain swaps to generate a set of chimeric ligands (II-V, VII-X in *Figure 1*) for stimulation of N1 and N2-expressing cells in our co-culture assay. Like wild-type ligands, chimeric ligands were expressed from single copy transgene integrations in the *Hprt* locus of murine ES cells. All chimeric proteins were expressed and present on the cell surface (*Supplementary file 1*), but expression levels varied among the chimeras (*Supplementary file 2*) despite integration into the *Hprt* locus by homologous recombination. We thus analyzed receptor selectivity of the chimeras in stimulating N1 and N2 responses using the co-culture assay by determining the N1/N2 response ratio. Stimulation with DLL1 gives a N1/N2 response ratio of approximately 1, DLL4 of ~20 (*Figure 5A*; numerical values used for calculations in *Figure 5—source data 1*, *Figure 5—source data 2*, *Figure 5—source data 3*; graphical representations of the relative luciferase activities of the ligands are shown in *Figure 5—figure supplement 1*). Strikingly, chimeras which retain the full ectodomain or the MNNL-EGF3 region of DLL4 have a N1/N2 stimulation ratio of approximately 20 similar to DLL4, whereas chimeras that retain the ectodomain, or MNNL-EGF3 region of DLL1 have a stimulation ratio of between one and two, resembling DLL1 (*Figure 5A*). These results indicate that the differences in activation potential of DLL4 and DLL1 toward N1 and N2 are encoded in the N-terminal part of the protein, encompassed by MNNL-EGF3. When chimeras include the MNNL-EGF2 or MNNL-DSL region of one ligand and the remainder of the other, the N1/N2 stimulation ratios of the chimeric pairs are equivalent (*Figure 5A*), indicating that the third EGF-like repeat makes an important contribution to receptor selectivity.

## Regions outside of the MNNL-DSL contact interface contribute to the functional difference of DLL1 and DLL4 in vitro and in vivo

To analyze to what extent the amino acids that contact N1 in the binding interfaces of the MNNL and DSL domains might contribute to the different activity of DLL1 and DLL4 toward N1 and N2 we reciprocally exchanged these amino acids (XI-XII in *Figure 1*; *Figure 5C* and *Figure 1—figure supplement 1B*) based on alignments of the DLL4 (*Luca et al., 2015*) and DLL1 (*Kershaw et al., 2015*) structures (*Figure 5B*). Western blot analyses of cell lysates and cell surface biotinylation and immunoprecipitation showed that all variants were present on the cell surface (*Supplementary File 1*). The N1/N2 response ratios show that swapping the contact residues of DLL4 onto DLL1 do not substantially affect the activation ratio when compared to DLL1 itself, indicating that the differences between DLL1 and DLL4 in N1/N2 selectivity cannot simply be accounted for by interfacial residues in the MNNL-DSL region (*Figure 5D*; numerical values used for calculations are in *Figure 5—source data 1*, *Figure 5—source data 2*, *Figure 5—source data 3*). Similarly, replacement of the DLL4 contact residues by the analogous residues of DLL1 slightly reduces the mean N1/N2 activation ratio (to ~13), but does not collapse the ratio to 1 (*Figure 5D*), again strongly suggesting that residues outside of the MNNL-DSL contact interface contribute to the relative N1 selectivity of DLL4. These results are consistent with 1) the domain swap data, which argue that discrimination between DLL1 and DLL4 depends on the EGF repeats as well as on the MNNL-DSL region, and 2) the prior observation that variants of DLL4 selected for high N1 affinity accumulate mutations in the protein core, but not in the binding interface (*Luca et al., 2015*). Swapping the contact residues of DLL1 onto DLL4 did not reduce the binding affinity of DLL4 for N1 (D4contD1 $K_D$ = 0.327 ± 0.036 μM; *Figure 1—figure supplement 2Ac*), fully consistent with the interpretation that the protein core of DLL4 contributes to N1 binding affinity, likely by influencing the fraction of molecules in a binding-active conformation. Although swapping the contact residues of DLL4 onto DLL1 increased binding affinity for N1 (D1contD4 $K_D$ = 0.326 ± 0.044 μM; *Figure 1—figure supplement 2Ad*), the substitution did not substantially change the N1/N2 activation ratio, indicating that binding affinity for N1 is not the only influence on the selectivity of the two ligands for N1 or N2.

The DLL4 MNNL domain contains three N-glycosylation sites, one of which (N109) is conserved from amphibian to mammalian DLL4 ligands but absent in DLL1. This residue resides adjacent to the contact amino acid F110 (*Figure 1—figure supplement 3A*). We confirmed that DLL4 can actually be N-glycosylated at this site (*Figure 1—figure supplement 3C*) and tested whether N109-glycosylation contributes to DLL4 activity and selectivity by mutating N109 to G (XIII in *Figure 1*), which is the amino acid present in DLL1 in the equivalent position (G112). D4$^{N109G}$ had no effect on the relative activation potential of DLL4 for N1 versus N2 (*Figure 5D*), and its affinity for N1 was not altered ($K_D$ = 0.341 ± 0.015 μM; *Figure 1—figure supplement 2Ae*), indicating that N-glycosylation at this site does not significantly modulate N1 binding or contribute to the relative selectivity of DLL4 towards N1 and N2.

To test whether the contact amino acids of DLL1 and DLL4 contribute to their functional divergence in vivo we generated a mouse line expressing D1contD4 (XI in *Figure 1*) instead of wild type DLL1 using our "mini-gene" knock-in strategy (*Figure 3A*). Heterozygous mice carrying this allele (*Dll1$^{D1contD4ki}$*) are indistinguishable from wild type. Homozygous mutants obtained from heterozygous matings at the expected Mendelian ratio (6/27) were viable and fertile, and indistinguishable from wild type and *Dll1$^{Dll1ki/Dll1ki}$* controls (*Figure 6A*). *Uncx* was expressed in regular pattern in the caudal halves of the somites of homozygous embryos (*Figure 6Cd,d'*), consistent with only subtle abnormalities of individual vertebral bodies in the lower thoracic region of *Dll1$^{D1contD4ki/D1contD4ki}$* fetuses (*Figure 6D*; n = 3/4) indicating that the contact amino acids and different binding affinities are not a major discriminating feature of the two ligands in vivo.

## Discussion

DLL1 and DLL4 have context-dependent redundant and divergent functions, but the bases for these differences are unclear. Here, using systematic domain exchanges and mutation of contact amino acids in the MNNL and DSL domains of DLL1 and DLL4, cell-based and biochemical assays, and transgenic mice we show that (1) DLL1 and DLL4 differ significantly in their potential to activate N1 and N2 and this difference is encoded in the ligand ectodomains, (2) regions outside the known

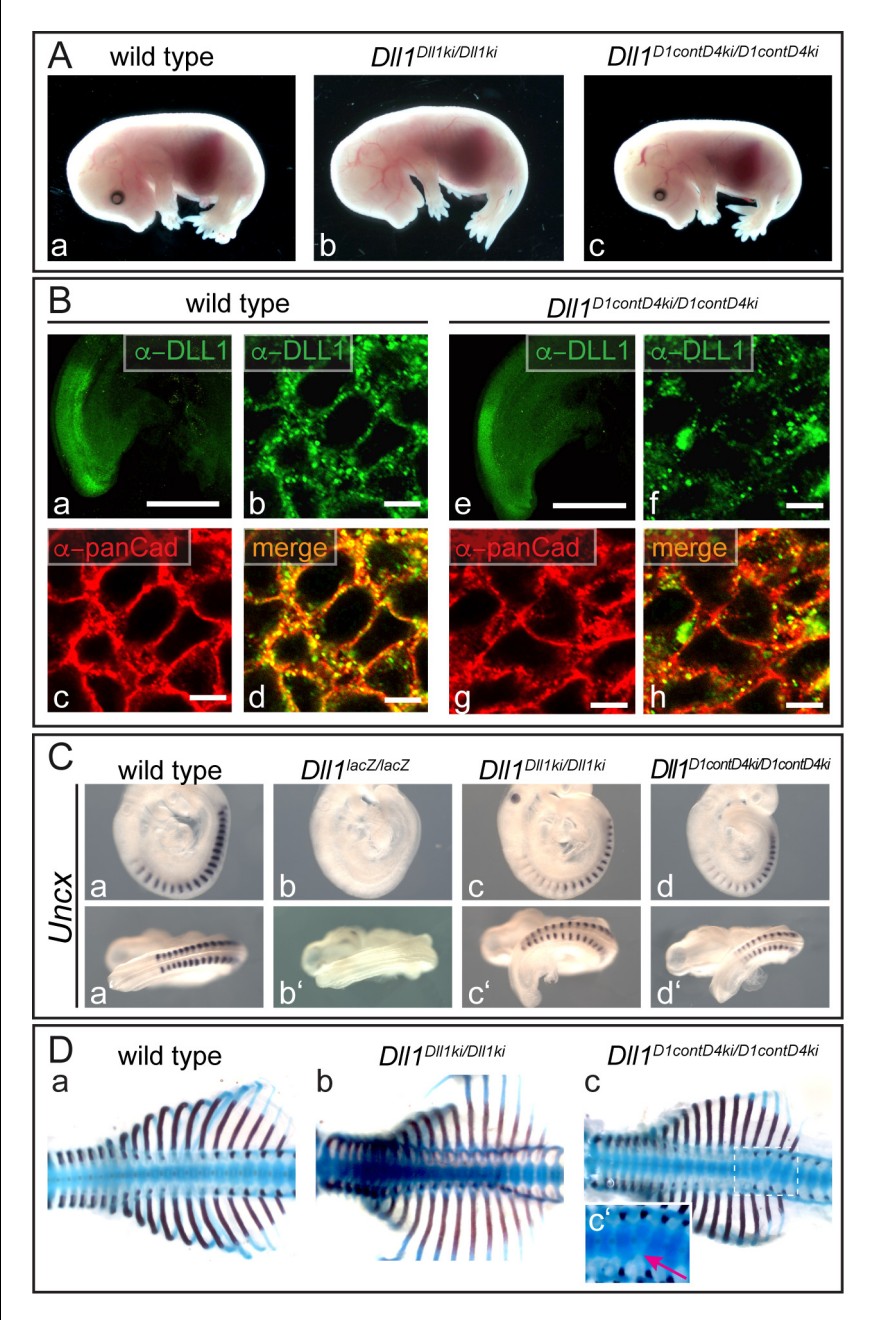

**Figure 6.** DLL1 carrying the DLL4 contact amino acids in the MNNL and DSL domains is a functional DLL1 ligand in vivo. (**A**) E15.5 *Dll1^D1contD4ki/D1contD4ki^* (c; n = 12) fetuses are indistinguishable from wild type (a; n = 19) and *Dll1^Dll1ki/Dll1ki^* (b; n = 3) controls. (**B**) D1contD4 co-localizes with pan-Cadherin (panCad) at the cell surface of *Dll1^D1contD4ki/D1contD4ki^* PSM cells (e-h; n ≥ 3); Scale bars: a, e = 500 µm; b-d, f-h = 5 µm. (**C**) Whole mount in situ hybridization showing that D1contD4 induces normal *Uncx* expression during somitogenesis (d,d'; n ≥ 5). (**D**) Skeletal preparations of *Dll1^D1contD4ki/D1contD4ki^* E15.5 fetuses showing minor defects of single vertebrae in the lower thoracic region (c,c'; n = 3/4).

DOI: https://doi.org/10.7554/eLife.40045.023

contact interface contribute to context-dependent ligand function, and (3) the contact amino acids are not the sole or primary determinant of this discrimination between the two receptors.

Analysis of our transgenic mice expressing D1*ECD*_D4*ICD* or D4*ECD*_D1*ICD* indicate a critical role of the ECD for the function of DLL1 during somite patterning in vivo. This resembles intrinsic

functional differences that reside in the extracellular domains of mN1 and mN2 during kidney development (*Liu et al., 2013*), whereas the N1 and N2 ICDs appear to be functionally equivalent in various developmental contexts (*Liu et al., 2015*). Functional equivalence of DLL1's and DLL4's ICDs in vivo is further supported by the rescue of the skeletal muscle phenotype in our D1*N-E3*_D4 knock-in mice, which harbor the DLL4 ICD. In this developmental context even domains C-terminal to EGF3 of DLL1 appear to be interchangeable. Analyses of the ECD/ICD domain swaps in the cell-based assay also suggest that the discriminatory potential of the ligands tracks with the ECD, and not with the ICD, even though the ICD appears to affect the strength and/or dynamics of the signal in co-culture assays where ligand and receptor expression is enforced in vitro (*Nandagopal et al., 2018*). Additional sources of complexity in vivo, like the stronger cis-inhibitory potential of the DLL4 ECD on Notch signaling (*Preuße et al., 2015*), or cyclic modulation of Notch by LFNG in the paraxial mesoderm, or different interactions with lipids (*Suckling et al., 2017*) might account for the resistance to loss of function phenotypes from ligand ICD swaps in vivo.

EGF-like repeats 11 and 12 of mouse N1 and N2 are highly similar (56/83 residues identical, 14 similar amino acids), and 13 of the 17 amino acid residues at the DLL4-binding interface are identical. Moreover, the x-ray structures of the EGF11-13 fragments of human N1 and N2 adopt a very similar arrangement (*Suckling et al., 2017*). Nevertheless, DLL1 and DLL4 exhibit a "discrimination potential" of ~20 fold in terms of receptor response in culture assays, suggesting that either the few different contact amino acids in EGF 11 and 12 of N1 and N2 have a significant impact or interactions of DLL1 and DLL4 with N1 and N2 are not limited to the MNNL and DSL interfaces with receptor EGF repeats 11 and 12. Domain swaps carried out here show that the region responsible for this receptor discrimination maps to the MNNL-EGF3 region (*Figure 5*). These findings are consistent with previous work uncovering the requirement of EGF repeats 1–3 of the DLL ligands for NOTCH1 activation, the importance of this region in the binding of Serrate family ligands to Notch receptors and in Serrate/Jagged-induced signaling, and the importance of EGF repeats 8–10 of NOTCH1 for signal activation by DLL ligands (*Shimizu et al., 1999*; *Cordle et al., 2008*; *Yamamoto et al., 2012*; *Andrawes et al., 2013*; *Schuster-Gossler et al., 2016*; *Luca et al., 2017*; *Liu et al., 2017*). Together, this body of work suggests that interactions of the N-terminal EGF repeats of the DLL ligands with EGF repeats 8–10 of Notch also contribute to recognition and impart discriminatory potential. The D1*N-E3*_D4 knock-in mice also point a functional role for domains outside the known binding interface, since this chimeric ligand does not substitute fully for DLL1 in vivo during somite patterning despite harboring the MNNL and DSL domains and EGF1-3 of DLL1, supporting context-dependent contributions of additional C-terminal EGF repeats observed previously in mice (*Schuster-Gossler et al., 2016*).

Remarkably, the exchange of the contact amino acids in DLL1 with those of DLL4 in the D1contD4 protein does not alter receptor selectivity in cultured cells even though these changes increase N1 binding affinity. This result suggests that receptor selectivity of DLL1 and DLL4 is not determined exclusively by the differences in binding strength. The D1contD4 chimera even substitutes almost completely for DLL1 function in mice during somite patterning, which is highly sensitive to altered Notch signaling (*Schuster-Gossler et al., 2009*) and therefore a suitable in vivo read out to detect even minor differences of Notch ligand function. Together, our results also favor the conclusion that the contact amino acids in the MNNL and DSL domains do not make the dominant contributions to the functional divergence of DLL1 and DLL4 in vivo, suggesting instead that differences in the domain cores, and/or contacts outside of the known DLL4-NOTCH1 interface, are the factors that most contribute to this functional divergence.

## Materials and methods

### Key resources table

| Reagent type (species) or resource | Designation | Source or reference | Identifiers | Additional information |
|---|---|---|---|---|
| Gene (*Mus musculus*) | DLL1 | MGI:104659; NCBI Gene: 13388 | | |

*Continued on next page*

*Continued*

| Reagent type (species) or resource | Designation | Source or reference | Identifiers | Additional information |
|---|---|---|---|---|
| Gene (*Mus musculus*) | DLL4 | MGI:1859388; NCBI Gene: 54485 | | |
| Strain, strain background (Mus musculus) | CD1 | Charles River Laboratories | | |
| Strain, strain background (Mus musculus) | 129Sv/CD1 hybrids | own colony | | |
| Genetic reagent (*Mus musculus*) | *Dll1$^{lacZ}$* | PMID: 9109488; DOI: 10.1038/386717a0 | RRID:MGI:5780046 | |
| Genetic reagent (*Mus musculus*) | *Dll1$^{loxP}$* | PMID: 15146182; DOI: 10.1038/ni1075 | RRID:MGI:5431505 | |
| Genetic reagent (*Mus musculus*) | T(s):Cre | PMID: 18708576; PMCID: PMC2518812; DOI: 10.1101/gad.480408 | MGI:3811072 | |
| Genetic reagent (*Mus musculus*) | ZP3:Cre | PMID: 10686600 | MGI:2176187 | |
| Genetic reagent (*Mus musculus*) | *Dll1$^{Dll1ki}$* | PMID: 26801181; PMCID: PMC4788113; DOI: 10.1534/genetics.115.184515 | MGI:5790945 | |
| Genetic reagent (*Mus musculus*) | *Dll1$^{Dll4ki}$* | PMID: 26114479; PMCID: PMC4482573; DOI: 10.1371/journal.pgen.1005328 | MGI:5779556 | |
| Genetic reagent (*Mus musculus*) | *Dll1$^{D1N-E3\_D4ki}$* | this paper | | mini gene insertion in the *Dll1* locus |
| Genetic reagent (*Mus musculus*) | *Dll1$^{D1contD4ki}$* | this paper | | mini gene insertion in the *Dll* locus |
| Genetic reagent (*Mus musculus*) | *Hprt$^{Dll1ECD\_Dll4ICD}$* | this paper | | inducible insertion into *Hprt* locus |
| Genetic reagent (*Mus musculus*) | *Hprt$^{Dll4ECD\_Dll1ICD}$* | this paper | | inducible insertion into *Hprt* locus |
| Cell line (*Mus musculus*) | E14TG2a | PMID: 26114479; PMCID: PMC4482573; DOI: 10.1371/journal.pgen.1005328 | | |
| Cell line (*Mus musculus*) | 129Sv/cast | PMID: 26114479; PMCID: PMC4482573; DOI: 10.1371/journal.pgen.1005328 | | |
| Cell line (*Homo sapiens*) | HeLaN1 | PMID: 9653148; DOI: 10.1073/pnas.95.14.8108 | | |
| Cell line (*Mus musculus*) | pMP8.CAG-DLL1 | PMID: 26801181; PMCID: PMC4788113; DOI: 10.1534/genetics.115.184515 | | |
| Cell line (*Mus musculus*) | E14rep | PMID: 26801181; PMCID: PMC4788113; DOI: 10.1534/genetics.115.184515 | | |

*Continued on next page*

*Continued*

| Reagent type (species) or resource | Designation | Source or reference | Identifiers | Additional information |
|---|---|---|---|---|
| Cell line (*Mus musculus*) | N1rep | PMID: 26801181; PMCID: PMC4788113; DOI: 10.1534/genetics. 115.184515 | | |
| Cell line (*Mus musculus*) | N2rep | PMID: 26801181; PMCID: PMC4788113; DOI: 10.1534/genetics. 115.184515 | | |
| Bacterial strain (*E. coli*) | SW106 | PMID:15731329 | | |
| Transfected construct (*Mus musculus*) | pMP8.CAG-DLL4 | this paper | | progenitor: pMP8.CAG |
| Transfected construct (*Mus musculus*) | pMP8.CAG-D1ECD _D4ICD | this paper | | progenitor: pMP8.CAG |
| Transfected construct (*Mus musculus*) | pMP8.CAG-D1N-E3_D4 | this paper | | progenitor: pMP8.CAG |
| Transfected construct (*Mus musculus*) | pMP8.CAG-D1N-E2_D4 | this paper | | progenitor: pMP8.CAG |
| Transfected construct (*Mus musculus*) | pMP8.CAG-D1N-D_D4 | this paper | | progenitor: pMP8.CAG |
| Transfected construct (*Mus musculus*) | pMP8.CAG-D4ECD_D1ICD | this paper | | progenitor: pMP8.CAG |
| Transfected construct (*Mus musculus*) | pMP8.CAG-D4N-E3_D1 | this paper | | progenitor: pMP8.CAG |
| Transfected construct (*Mus musculus*) | pMP8.CAG-D4N-E2_D1 | this paper | | progenitor: pMP8.CAG |
| Transfected construct (*Mus musculus*) | pMP8.CAG-D4N-D_D1 | this paper | | progenitor: pMP8.CAG |
| Transfected construct (*Mus musculus*) | pMP8.CAG-D1contD4 | this paper | | progenitor: pMP8.CAG |
| Transfected construct (*Mus musculus*) | pMP8.CAG-D4contD1 | this paper | | progenitor: pMP8.CAG |
| Transfected construct (*Mus musculus*) | pMP8.CAG-D4N109G | this paper | | progenitor: pMP8.CAG |
| Transfected construct (*Mus musculus*) | pMP8.CAG-Stop-D1ECD_D4ICD | this paper | | progenitor: pMP8.CAG |
| Transfected construct (*Mus musculus*) | pMP8.CAG-Stop-D4ECD_D1ICD | this paper | | progenitor: pMP8.CAG |
| Transfected construct (*Mus musculus*) | D1N-E3_D4-targeting | this paper | | based on Dll1Dll1ki targeting |
| Transfected construct (*Mus musculus*) | D1contD4-targeting | this paper | | based on Dll1Dll1ki targeting |
| Transfected construct (*Mus musculus*) | pLexM-Avi-His | this paper | | progenitor: pLexM |
| Transfected construct (*Mus musculus*) | pLexM-D1N-E5-Avi-His | this paper | | progenitor: pLexM |
| Transfected construct (*Mus musculus*) | pLexM-D4N-E5-Avi-His | this paper | | progenitor: pLexM |
| Transfected construct (*Mus musculus*) | pLexM-D1N-E3_D4-E5-Avi-His | this paper | | progenitor: pLexM |

*Continued on next page*

Continued

| Reagent type (species) or resource | Designation | Source or reference | Identifiers | Additional information |
|---|---|---|---|---|
| Transfected construct (*Mus musculus*) | pLexM-D4N-E3_D1-E5-Avi-His | this paper | | progenitor: pLexM |
| Transfected construct (*Mus musculus*) | pLexM-D1contD4-E5-Avi-His | this paper | | progenitor: pLexM |
| Transfected construct (*Mus musculus*) | pLexM-D4contD1-E5-Avi-His | this paper | | progenitor: pLexM |
| Transfected construct (*Mus musculus*) | pLexM-D4N109G-E5-Avi-His | this paper | | progenitor: pLexM |
| Antibody | Rat anti-DLL1 | PMID: 17664336; PMCID: PMC2064846; DOI: 10.1083/jcb.200702009 | (1F9, rat monoclonal) | 1:50 (IF) |
| Antibody | Goat anti-DLL4 | R and D Systems | Cat. #AF1389 RRID:AB_354770 | 1:50 (IF) |
| Antibody | Mouse anti-panCadherin | Sigma-Aldrich | Cat. #C1821 RRID:AB_476826 | 1:250 (IF) |
| Antibody | Donkey anti-mouse Alexa 555 | Invitrogen | Cat. #A-31570 RRID:AB_2536180 | 1:100 (IF) |
| Antibody | Donkey anti-goat Alexa 488 | Invitrogen | Cat. #A-11055 RRID:AB_2534102 | 1:100 (IF) |
| Antibody | Donkey anti-rat Alexa 488 | Invitrogen | Cat. #A-21208 RRID:AB_2535794 | 1:100 (IF) |
| Antibody | Anti-FLAG-Peroxidase (HRP) | Sigma-Aldrich | (M2 mouse, monoclonal purified) Cat. #A8592 RRID:AB_439702 | 1:10 000 (WB) |
| Antibody | Mouse anti-β-Tubulin | Sigma-Aldrich | Cat. #T7816 RRID:AB_261770 | 1:500 000; 1:1 000 000 (WB) |
| Antibody | Anti-mouse HRP | Amersham | Cat. #NA931 RRID:AB_772210 | 1:10 000 (WB) |
| Antibody | MHC (Myosin Heavy Chain) | Sigma-Aldrich | Cat. #M4276 RRID:AB_477190 | 1:250 (IHC) |
| Antibody | Anti-DIG AP fab fragment | Roche | Cat. #1093274 | 1: 5 000 (ISH) |
| Antibody | Anti-mouse biotinylated (BA9200/goat) | Vector Laboratories | Cat. #BA-9200 RRID:AB_2336171 | 1:200 (IHC) |
| Commercial assay or kit | Luciferase Cell Culture Lysis 5X Reagent | Promega | Cat. #E1531 | |
| Commercial assay or kit | Luciferase Assay Reagent | Promega | Cat. #E1483 | |
| Commercial assay or kit | SuperScript IV Reverse Transcriptase | Invitrogen | Cat. #18090050 | |
| Commercial assay or kit | Expand High-Fidelity PCR system | Roche | Cat. #04743733001 | |
| Commercial assay or kit | Tri-Reagent | Sigma-Aldrich | Cat. #T9424 | |
| Chemical compound, drug | Sulfo-NHS-LC-Biotin | Thermo | Cat. #21335 | |
| Chemical compound, drug | Pierce NeutrAvidin Agarose | Thermo | Cat. #29200 | |

*Continued on next page*

*Continued*

| Reagent type (species) or resource | Designation | Source or reference | Identifiers | Additional information |
|---|---|---|---|---|
| Chemical compound, drug | cOmplete, Mini, EDTA-free Proteinase Inhibitor Cocktail | Roche | Cat. #04693159001 | |
| Chemical compound, drug | BM-Purple AP substrate Roche | Sigma-Aldrich | Cat. #11442074001 | |
| Chemical compound, drug | G418 | Biochrom | Cat. #291–25 | 125 µg/ml |
| Chemical compound, drug | HAT | Gibco | Cat. #31062–037 | 1:300 |
| Chemical compound, drug | HT | Gibco | Cat. #11067030 | 1:100 |
| Chemical compound, drug | Tunicamycin | Sigma-Aldrich | Cat. #T7765 | 1 µg/ml |
| Chemical compound, drug | Alcian blue | Sigma-Aldrich | Cat. #A5268 | 5% working solution |
| Chemical compound, drug | Alizarin red | Sigma-Aldrich | Cat. #A5533 | 5% working solution |
| Other | WesternBright Quantum HRP substrate | Advansta | Cat. #12042-D20 | as recommended by the manufacturer |
| Other | Amersham ECL Detection Reagent | GE Healthcare Life Sciences | Cat. #RPN2106 | as recommended by the manufacturer |
| Sequence-based reagent | DLL1 wt For | other | NA | 5'-CTGAAGCGACCT GGCCCTGATAGCAC-3' |
| Sequence-based reagent | DLL1 wt Rev | other | NA | 5'-GGAGCTCCAGA CCTGCGCGGG-3' |
| Sequence-based reagent | *Dll1$^{lacZ}$* For | other | NA | 5'-ATCCCTGGGT CTTTGAAGAAG-3' |
| Sequence-based reagent | *Dll1$^{lacZ}$* Rev | other | NA | 5'-TGTGAGCGAGTA ACAACCCGTCGGATT-3' |
| Sequence-based reagent | *Dll1$^{Dll1ki}$* For | other | NA | 5'-GGTTTGGGGAT CCATAACTTCG-3' |
| Sequence-based reagent | *Dll1$^{Dll1ki}$* Rev | other | NA | 5'-GCCAGTCAGTTC CCAGTAAGAAGTC-3' |
| Sequence-based reagent | *Dll1$^{Dll4ki}$* For | other | NA | 5'-AAGGACAACC TAATCCCTGCCG-3' |
| Sequence-based reagent | *Dll1$^{Dll4ki}$* Rev | other | NA | 5'-TGCCACATCG CTTCCATCTTAC-3' |
| Sequence-based reagent | *Dll1$^{loxP}$* For | other | NA | 5'-GCATTTCTCAC ACACCTC-3' |
| Sequence-based reagent | *Dll1$^{loxP}$* Rev | other | NA | 5'-GAGAGTACTT GATGGAGCAAG-3' |
| Sequence-based reagent | *T(s):Cre* For | other | NA | 5'-AATCTTTGG GCTCCGCAGAG-3' |
| Sequence-based reagent | *T(s):Cre* Rev | other | NA | 5'-ACGTTCACCGGC ATCAACG-3' |
| Sequence-based reagent | *ZP3:Cre* For | other | NA | 5'-GCCTGCATTACC GGTCGATGCAACGA-3' |
| Sequence-based reagent | *ZP3:Cre* Rev | other | NA | 5'-GTGGCAGATGGC GCGGCAACACCATT-3' |
| Sequence-based reagent | *Hprt-CAGD1ECD_D4ICD + neo* For | this paper | NA | 5'-CCTAGCCCCTGCA AGAACGGAGC-3' |

*Continued on next page*

*Continued*

| Reagent type (species) or resource | Designation | Source or reference | Identifiers | Additional information |
|---|---|---|---|---|
| Sequence-based reagent | *Hprt-CAGD1ECD_D4ICD + neo* Rev | this paper | NA | 5′-TTGCCACAATTG GACTTGTC-3′ |
| Sequence-based reagent | *Hprt-CAGD4ECD_D1ICD + neo* For | this paper | NA | 5′-CACTGTGAGCAT AGTACC TTGAC-3′ |
| Sequence-based reagent | *Hprt-CAGD4ECD_D1ICD + neo* Rev | this paper | NA | 5′-CATGGTTTCTGTCT CTCCCCCACAGGG-3′ |
| Sequence-based reagent | *Hprt*$^{D1ECD\_D4ICDrec}$ and *Hprt*$^{D4ECD\_D1ICDrec}$ For (activated allele) | this paper | NA | 5′-ACATGGCCGTCATC AAAGAG-3′ |
| Sequence-based reagent | *Hprt*$^{D1ECD\_D4ICDrec}$ and *Hprt*$^{D4ECD\_D1ICDrec}$ Rev (activated allele) | this paper | NA | 5′-GGGCAACAGAGA AATATCCTGTCTC-3′ |
| Sequence-based reagent | *Dll1*$^{D1N-E3\_D4ki}$ For | this paper | NA | 5′-CTGTCTGCCAGG GTGTGATGACCAAC-3′ |
| Sequence-based reagent | *Dll1*$^{D1N-E3\_D4ki}$ Rev | this paper | NA | 5′-ATCGCTGATG TGCAGTTCACA-3′ |
| Sequence-based reagent | *Dll1*$^{D1N-E3\_D4ki}$ For | this paper | NA | 5′-TGCAGGAG TTCGTCAACAAG-3′ |
| Sequence-based reagent | *Dll1*$^{D1N-E3\_D4ki}$ Rev | this paper | NA | 5′-ATAGTGGCC AAAGTGGTCATC CCGAGGCTT-3′ |
| Sequence-based reagent | Y-Chromosome For | other | NA | 5′-CTGGAGCTCT ACAGTGATGA-3′ |
| Sequence-based reagent | Y-Chromosome Rev | other | NA | 5′-CAGTTACCAA TCAACACATCAC-3′ |
| Sequence-based reagent | *mNotch1* For | other | NA | 5′-TAGGTGCTC TTGCGTCACTTGG-3′ |
| Sequence-based reagent | *mNotch1* Rev | other | NA | 5′-TCTCCCCACT CGTTCTGATTGTC-3′ |
| Sequence-based reagent | *hNOTCH1* For | PMID: 22002304; DOI: 10.1038/onc.2011.467 | NA | 5′-TCCACCAG TTTGAATGGTCA-3′ |
| Sequence-based reagent | *hNOTCH1* Rev | PMID: 22002304; DOI: 10.1038/onc.2011.467 | NA | 5′-AGCTCATCA TCTGGGACAGG-3′ |
| Sequence-based reagent | *hNOTCH2* For | this paper | NA | 5′-CAACCGCCA GTGTGTTCAAG-3′ |
| Sequence-based reagent | *hNOTCH2* Rev | this paper | NA | 5′-GAGCCATG CTTACGCTTTCG-3′ |
| Sequence-based reagent | *hNOTCH3* For | PMID: 16327489; PMCID: PMC1409885 | NA | 5′-AGATTCTCA TCCGAAACCGCTCTA-3′ |
| Sequence-based reagent | *hNOTCH3* Rev | PMID: 16327489; PMCID: PMC1409885 | NA | 5′-GGGGTCTC CTCCTTGCTATCCTG-3′ |
| Sequence-based reagent | *hGAPDH* For | PMID: 22002304; DOI: 10.1038/onc.2011.467 | NA | 5′-GAGTCAACG GATTTGGTCGT-3′ |
| Sequence-based reagent | *hGAPDH* Rev | PMID: 22002304; DOI: 10.1038/onc.2011.467 | NA | 5′-TTGATTTTGG AGGGATCTCG-3′ |
| Sequence-based reagent | Forward primer - correct integration into *Hprt* locus | other | NA | 5′-GGGAACCTGTT AGAAAAAAGA AACTATGAAGAAC-3′ |

*Continued on next page*

*Continued*

| Reagent type (species) or resource | Designation | Source or reference | Identifiers | Additional information |
|---|---|---|---|---|
| Sequence-based reagent | Reverse primer - correct integration into *Hprt* locus | other | NA | 5'-GGCTATGAACTAATG GACCCCG-3' |
| Sequence-based reagent | Forward primer - correct integration into *Dll1* locus | other | NA | 5'-TGTCACGT CCTGCACGACG-3' |
| Sequence-based reagent | Reverse primer - correct integration into *Dll1* locus | other | NA | 5'-GGTATCGGA TGCACTCATCGC-3' |
| Sequence-based reagent | guideA-For | this work, according to http://crispr.mit.edu/ | NA | 5'-GGCAGCGGG CAGCTCCGGAT-3' |
| Sequence-based reagent | guideB-Rev | this work, according to http://crispr.mit.edu/ | NA | 5'-GCTCTCGGG GTCGTCGCTGC-3' |
| Recombinant DNA reagent | Uncx-probe (plasmid) | DOI 10.1007/s004270050120 | | |
| Recombinant DNA reagent | pLexM (plasmid) | DOI 10.1074/jbc.M113.454850 | | |
| Recombinant DNA reagent | Cas9 D10A nickase (plasmid) | DOI 10.1126/science.1231143 | Addgene #42335 | |
| Recombinant DNA reagent | *Dll1* 5' SB probe | PMID: 26801181; PMCID: PMC4788113; DOI: 10.1534/genetics.115.184515 | | 5' probe: a 316 bp BamHI/AvaII fragment 3.8 kb upstream of *Dll1* exon 1 |
| Recombinant DNA reagent | *Dll1* 3' SB probe | PMID: 26801181; PMCID: PMC4788113; DOI: 10.1534/genetics.115.184515 | | 3' probe: a 528 bp PCR fragment in *Dll1* intron five obtained with primers CCTGTGAGACTTTCTA CGTTGCTC/CACAACCATGTCA CCTTCTAGATTC |
| Software, algorithm | ImageJ; FIJI | | RRID:SCR_003070 | ISAC Manager |
| Software, algorithm | Prism | GraphPad | RRID:SCR_002798 | |
| Software, algorithm | Olympus | Olympus FLUOVI EW FV1000 | RRID:SCR_014215 | |

## Generation and husbandry of transgenic mice

### Ethics statement

All animal experiments were performed according to the German rules and regulations (Tierschutz-gesetz) and approved by the ethics committee of Lower Saxony for care and use of laboratory animals LAVES (Niedersächsisches Landesamt für Verbraucherschutz und Lebensmittelsicherheit; refs.: 33.12-42505-04-13/1314 and 33.14-42505-04-13/1293). Mice were housed in the central animal facility of Hannover Medical School (ZTL) and were maintained as approved by the responsible Veterinary Officer of the City of Hannover. Animal welfare was supervised and approved by the Institutional Animal Welfare Officer (Tierschutzbeauftragter).

### Mouse strains

Wild type mice were CD1 and 129Sv/CD1 hybrids; *Dll1^lacZ* (*Hrabě de Angelis et al., 1997*), *Dll1^loxP* (*Hozumi et al., 2004*), T(s):Cre (*Feller et al., 2008*) and ZP3:Cre (*de Vries et al., 2000*), *Dll1^Dll1ki* (*Schuster-Gossler et al., 2016*), and *Dll1^Dll4ki* (*Preuße et al., 2015*) were described previously.

### Generation of transgenic mice

Mice allowing for inducible expression of chimeric ligands were generated by morula injection of E14TG2a ES cells carrying the expression construct in the *Hprt* locus. E14TG2a cells were

electroporated with linearized targeting constructs and correct integrations were identified by HAT selection and validated by long-range PCR using primers: For/Rev: GGGAACCTGTTAGAAAAAAA-GAAACTATGAAGAAC/GGCTATGAACTAATGACCCCG.

$Dll1^{D1N-E3\_D4ki}$ and $Dll1^{D1contD4ki}$ mice were generated with 129Sv/cast ES cells. ES cells were electroporated with linearized targeting constructs, Cas9 D10A nickase (Addgene #42335; **Cong et al., 2013**) expression vector and guide RNAs targeting the first intron of *Dll1* to increase the frequency of homologous recombination (guideA-FOR: GGCAGCGGGCAGCTCCGGAT; guideB-REV: GCTC TCGGGGTCGTCGCTGC, according to http://crispr.mit.edu/ the pair score for A and B – 79, 0 off target pairs, and 0 genic OT pairs). G418 resistant clones were screened for targeted integrations by long-range PCR using primers: For/Rev TGTCACGTCCTGCACGACG/GGTATCGGATGCACTCA TCGC and correct targeting events verified by Southern blot analysis (5' probe: a 316 bp BamHI/ AvaII fragment 3.8 kb upstream of *Dll1* exon 1; 3' probe: a 528 bp PCR fragment in *Dll1* intron five obtained with primers CCTGTGAGACTTTCTACGTTGCTC/CACAACCATGTCACCTTCTAGATTC). The *neo$^r$* cassette was excised in the female germ line using ZP3:Cre mice.

## Genotyping of mice and embryos

Genomic DNA was isolated from ear or tail biopsies, yolk sacs or umbilical cords and used as template in PCRs with the following primer pairs: $Hprt^{Dll1ECD\_Dll4ICD}$ For/Rev CTGTCTGCCAGGGTGTGA TGACCAAC/CAGATTGTTCATGGCTTCCCT; $Hprt^{Dll4ECD\_Dll1ICD}$ For/Rev CACTGTGAGCATAGTACC TTGAC/CATGGTTTCTGTCTCTCCCCCACAGGG; $Hprt^{Dll1ECD\_Dll4ICDrec}$ or $Hprt^{Dll4ECD\_Dll1ICDrec}$ (activated alleles) For/Rev ACATGGCCGTCATCAAAGAG/GGGCAACAGAGAAATATCCTGTCTC; $Dll1^{loxP}$ For/Rev GCATTTCTCACACACCTC/GAGAGTACTTGATGGAGCAAG; T(s):Cre For/Rev AA TCTTTGGGCTCCGCAGAG/ACGTTCACCGGCATCAACG; ZP3:Cre For/Rev GCCTGCATTACCGG TCGATGCAACGA/GTGGCAGATGGCGCGGCAACACCATT; $Dll1^{wt}$ For/Rev CTGAAGCGACC TGGCCCTGATAGCAC/GGAGCTCCAGACCTGCGCGGG; $Dll1^{lacZ}$ For/Rev ATCCCTGGGTC TTTGAAGAAG/TGTGAGCGAGTAACAACCCGTCGGATT; $Dll1^{Dll4ki}$ For/Rev AAGGACAACCTAA TCCCTGCCG/TGCCACATCGCTTCCATCTTAC; $Dll1^{Dll1ki}$ For/Rev GGTTTGGGGATCCATAACTTCG/ GCCAGTCAGTTCCCAGTAAGAAGTC; $Dll1^{D1N-E3\_D4ki}$ For/Rev CTGTCTGCCAGGGTGTGATGAC-CAAC/ATCGCTGATGTGCAGTTCACA; $Dll1^{D1contD4ki}$ For/Rev TGCAGGAGTTCGTCAACAAG/ATAG TGGCCAAAGTGGTCATCCCGAGGCTT; Y-chromosome PCR For/Rev CTGGAGCTCTACAGTGA TGA/CAGTTACCAATCAACACATCAC

## Cloning of constructs

### *Hprt* targeting constructs for expression from single copy integrations in ES cells

cDNAs encoding Flag-tagged ligand proteins with exchanges of domains or individual amino acids in the extracellular domain of DLL1 and DLL4 were generated by standard cloning procedures using either synthesized gene fragments (II-IV, VII-IX, XI-XIII in **Figure 1**) or fragments obtained by restriction digests from *Dll1* and *Dll4* cDNA constructs (V, X in **Figure 1**). Tagged cDNAs were cloned into pMP8-CAG.Stop-shuttle as *Eco*RI/*Bam*HI or *Eco*RI/*Not*I fragments. The stop cassette was excised by Cre mediated recombination of the *loxP* sites in bacterial SW106 cells.

### *Hprt* targeting constructs for inducible expression in transgenic mice

D1*ECD*_D4*ICD* and D4*ECD*_D1*ICD* were generated by PCR amplification of the respective untagged cDNAs and subcloned into shuttle vector pSL1180dttomato containing the wild type and mutant *loxP* sites and iresdsRED. Subsequently, the fragments encoding the chimeric ligands fused to iresdsRED were cloned into pMP8-CAG.Stop (**Preuße et al., 2015**) using *Mlu*I and *Swa*1 restriction sites.

### Mini gene constructs for targeting the *Dll1* locus

$Dll1^{D1N-E3\_D4ki}$ and $Dll1^{D1contD4ki}$ targeting constructs were generated by standard cloning procedures based on the $Dll1^{Dll1ki}$ ($Dll1^{tm7.1Gos}$) or $Dll1^{Dll4ki}$ ($Dll1^{tm4.1Gos}$) targeting vectors (**Preuße et al., 2015**; **Schuster-Gossler et al., 2016**). First, the 3' DT cassette was removed by *Pme*I and *Aat*II digest and relegation of the blunt ended plasmid. EcoRI fragments containing the *Dll1* or *Dll4* coding sequences in the targeting vector lacking the 3' DT cassette were excised by *Eco*RI and cloned

into pCR-TOPO-XL. The wild type *Dll1* sequence was replaced in pCR-TOPO-XL by a D1contD4 cDNA, the *Dll4* sequence by D1*N*-E3_D4 cDNA. Fragments were ligated back into the targeting vectors as *EcoR*I fragments.

### Avi-His-tagged ligand fragments for protein expression and purification
For production and purification of proteins for binding assays (XIV-XVIII in *Figure 1*) fragments encompassing the N-terminus up to and including EGF5 were PCR amplified and cloned into pLexM-Avi-His vector (*Andrawes et al., 2013*) as *EcoR*I/*BamH*I fragments by standard procedures.

## Analysis of gene expression patterns and phenotypes

### Whole mount in situ hybridization
E9.5 embryos were collected in ice cold PBS and fixed in 4% formaldehyde/PBS over night at 4°C and dehydrated in methanol. In situ hybridization was performed according to standard procedures with digoxigenin labelled cDNA probes for *Uncx* (*Neidhardt et al., 1997*).

### Antibody staining
E18.5 embryos were collected in ice cold PBS, fixed in 4% formaldehyde/PBS over night at 4°C, dehydrated in methanol, ethanol, and 2-propanol. Hind limbs were paraffin embedded and 10 µm transverse sections stained for Myosin Heavy Chain (MHC).

### Whole mount immunofluorescence
E9.5 embryos were collected in ice cold PBS, fixed in 4% formaldehyde in PBS and immunofluorescence staining was performed as described in *Bone et al. (2014)*. Used primary antibodies: anti-DLL1 (1F9; 1:50) (*Geffers et al., 2007*), anti-DLL4 (AF1389, R and D; 1:50), and anti-pan-Cadherin (C1821, Sigma; 1:250). Used secondary antibodies: Alexa488/555 conjugated antibodies (Invitrogen; 1:100). Images were taken using OLYMPUS FV1000.

### Skeletal preparations
E15.5 fetuses were collected in ice cold PBS and dehydrated in EtOH. Alcian blue and Alizarin red staining was performed using standard procedures (*Cordes et al., 2004*).

### Western blot analyses
Cells were lysed in 2x sample buffer (0.125M Tris pH 6.8; 4% SDS; 20% glycin; 5% β-mercaptoethanol; 0.025% bromphenol blue). Proteins were separated by SDS-PAGE and transferred onto Immobilon-P Transfer membranes (Millipore) by wet tank or SemiDry blotting. Membranes were blocked in 5% nonfat dried milk powder (AppliChem) in PBS/0.1% Tween20 and subsequently incubated in 5% nonfat dried milk powder containing primary antibodies. Used primary antibodies: anti-Flag HRP (mouse monoclonal; clone M2; Sigma; 1:10 000), anti-b-Tubulin I (Sigma; 1:500 000/1:1 000 000). Used secondary antibodies: anti-mouse HRP (Amersham; 1:10 000). For HRP detection ECL Western Blotting Detection Reagent (Amersham) and WesternBright Quantum (advansta) were used with Luminiscent Image Analyser LAS4000 (Fujifilm). ImageJ was used to quantify signals.

### RT-PCR
HeLaN1 cells were lysed in Tri-Reagent (Sigma) and RNA was isolated according to the manufacturer's instructions. Reverse transcription was performed using SuperScript IV (Invitrogen) according to the manufacturer's instructions. Primers used for RT-PCR analysis were: *mNotch1* For/Rev TAGG TGCTCTTGCGTCACTTGG/TCTCCCCACTCGTTCTGATTGTC; *hNOTCH1* For/Rev TCCACCAG TTTGAATGGTCA/AGCTCATCATCTGGGACAGG (*Ding et al., 2012*); *hNOTCH2* For/Rev CAACCGCCAGTGTGTTCAAG/GAGCCATGCTTACGCTTTCG; *hNOTCH3* For/Rev AGATTCTCA TCCGAAACCGCTCTA/GGGGTCTCCTCCTTGCTATCCTG (*Büchler et al., 2005*); *hGAPDH* For/Rev GAGTCAACGGATTTGGTCGT/TTGATTTTGGAGGGATCTCG (*Ding et al., 2012*).

## Southern blot analyses

Genomic DNA was isolated from ES cells, digested with BamHI overnight and separated on an 0.7% agarose gel. Blotting, crosslinking, hybridization, and signal detection were performed using Immobilon-Ny+ membrane (Millipore) according to the manufacturer's instructions.

## Cell culture experiments

### Culture of cells

Mouse E14TG2a and 129Sv/cast ES cells were cultured in DMEM (Invitrogen) cell culture medium supplemented with 15% FCS (Biochrom AG), Glutamax, Pen/Strep, Sodium Pyruvate, MEM Non-Essential Amino Acid Solution, β-mercaptoethanol, and leukemia inhibitory factor (LIF), HeLaN1 cells were cultured in DMEM (Invitrogen) cell culture medium supplemented with 10% FCS (Biochrom AG), Glutamax and Pen/Strep. All cell lines were tested negative for mycoplasma. No authentication of cell lines was performed.

### Generation of cells stably expressing ligand proteins

ES cells were electroporated with linearized pMP8 targeting vectors and selected with HAT (1:300; Gibco). Correct integration of the 5' homology arm in HAT resistant clones was verified with long-range PCR using following primers: For/Rev: GGGAACCTGTTAGAAAAAAAGAAACTATGAAGAAC/ GGCTATGAACTAATGACCCCG. Expression of proteins was verified using Western Blot analyses.

### Trans-activation assay

For in vitro cell co-culture assays ES cells were counted in PBS using LUNA-II (logos biosystems) and $9.25 \times 10^5$ ligand and $0.75 \times 10^5$ receptor expressing cells were plated on gelatin coated six well plate dishes. After 24 hr fresh medium was added. 48–52 hr after co-cultivation cells were washed once with PBS, lysed in 250 µl 1xCCLR (Luciferase Cell Culture Lysis Reagent, Promega), transferred into 1.5 ml tubes, and frozen at −80°C. For measurements lysates were thawed, vortexed, and briefly centrifuged. 20 µl aliquots of each lysate was measured with Luciferase Assay Reagent in duplicates or quadruplicates using GloMax-96 (Promega).

### Biotinylation assay

For determination of relative cell surface protein levels, cells were treated with Sulfo-NHS-LC-Biotin (Pierce; 0.25 mg/ml PBS supplemented with 1 mM $MgCl_2$ and 0.1 mM $CaCl_2$), quenched with 100 mM glycine in DMEM and lysed in lysis buffer supplemented with Complete Proteinase Inhibitor Cocktail Tablets (Roche). Biotinylated proteins were immunoprecipitated using NeutrAvidin beads (Thermo Scientific) and analyzed by Western blotting. For detailed information see (*Braune et al., 2014*; *Preuße et al., 2015*).

### Protein expression and purification

The cDNA for expression of the N1 fragment using the pLexM vector was described previously (*Andrawes et al., 2013*) and encodes the N1 signal sequence followed by EGF repeats 6–15 (amino acids 216–604), a biotinylation (avi) tag, and a His$_6$ tag. The cDNAs for expression of DLL1, DLL4, and all chimeric proteins extend from the N-terminus through EGF5. These proteins were also subcloned into pLexM as described (*Andrawes et al., 2013*). The N2(1–15)-Fc protein was purchased from R and D systems and used without further purification.

Expi293F cells maintained in Expi293 expression media were grown to cell density of $10^6$ cells/ml and then transiently transfected with *Dll1* ligand, *Dll4* ligand or *N1* DNA (1 mg/liter of cells) and FectoPro transfection reagent (Polyplus) at 1:1 DNA/FectoPro ratio. For biotinylation of Avi-tagged NOTCH1 protein, cells were co-transfected with biotin ligase (BirA) DNA as well as with DNA encoding Protein O-fucosyltransferase-1 (POFUT1), which enhances Notch folding and secretion.

Transfected cells were then cultured in FreeStyle293 media for 3–4 days to produce protein. The media was collected, separated from the cells by centrifugation and supplemented with 50 mM Tris buffer, pH 8.0. The resulting supernatant was bound to Ni-NTA beads (Qiagen) over a 3 hr incubation at 4° C. After a wash with ten column volumes of 50 mM Tris buffer, pH 8.0, containing 150 mM NaCl, 5 mM $CaCl_2$, and 20 mM Imidazole, bound protein was eluted with the same buffer supplemented with 250 mM Imidazole. Following elution, fractions containing the partially purified proteins

were concentrated and further purified by gel-filtration chromatography using a Superdex 200 column in 50 mM Tris, pH 8.0, containing 150 mM NaCl, and 5 mM $CaCl_2$. The quality of the resulting purified proteins was assessed using non-reducing SDS-PAGE. Pure fractions were pooled, flash frozen and stored at $-80°$ C. The efficiency of biotinylation was estimated by immunoprecipitation with streptavidin resin.

## Biolayer interferometry
Ligand binding affinities were quantified by biolayer interferometry using a BLItz instrument (Forte-Bio). For N1 binding, streptavidin biosensors were loaded with the biotinylated Notch1 fragment, equilibrated in buffer for 30 s, then dipped into ligand samples of varying concentration until equilibrium was observed. For N2 binding, protein A biosensors were used for the capture step. All ligand-receptor binding experiments were done in HBS-P buffer containing 0.005% surfactant P20, supplemented with 5 mM $CaCl_2$. Equilibrium binding curves were fitted with a one site - specific binding model using GraphPad Prism.

## Statistical analysis
Statistical analyses were done using Prism7 (GraphPad) as indicated in Figure legends.

## Acknowledgements
We thank Kristina Preusse for providing the cDNAs for D1N-E2_D4, D1N-D_D4, D4N-E2_D1, and D4N-D_D1 chimeric ligands and Patricia Delany-Heiken for excellent technical assistance.

## Additional information

### Funding

| Funder | Grant reference number | Author |
|---|---|---|
| Harvard Medical School | van Maanen Graduate fellowship | Sanchez M Jarrett |
| National Institutes of Health | R35-CA220340 | Stephen C Blacklow |
| Deutsche Forschungsgemeinschaft | GO 449/13-1 | Achim Gossler |
| Deutsche Forschungsgemeinschaft | REBIRTH | Achim Gossler |

The funders had no role in study design, data collection and interpretation, or the decision to submit the work for publication.

### Author contributions
Lena Tveriakhina, Formal analysis, Investigation, Visualization, Writing—original draft, Writing—review and editing; Karin Schuster-Gossler, Resources, Investigation, Writing—review and editing; Sanchez M Jarrett, Marie B Andrawes, Meike Rohrbach, Investigation, Writing—review and editing; Stephen C Blacklow, Achim Gossler, Conceptualization, Supervision, Funding acquisition, Visualization, Writing—original draft, Writing—review and editing

### Author ORCIDs
Achim Gossler (iD) http://orcid.org/0000-0002-9103-9116

### Ethics
Animal experimentation: All animal experiments were performed according to the German rules and regulations (Tierschutzgesetz) and approved by the ethics committee of Lower Saxony for care and use of laboratory animals LAVES (Niedersächsisches Landesamt für Verbraucherschutz und Lebensmittelsicherheit; refs.: 33.12-42502-04-13/1314 and 33.14-42502-04-13/1293). Mice were housed in the central animal facility of Hannover Medical School (ZTL) and were maintained as approved by

the responsible Veterinary Officer of the City of Hannover. Animal welfare was supervised and approved by the Institutional Animal Welfare Officer (Tierschutzbeauftragter).

## Decision letter and Author response

Decision letter https://doi.org/10.7554/eLife.40045.029
Author response https://doi.org/10.7554/eLife.40045.030

## Additional files

### Supplementary files

• Supplementary file 1. Relative cell surface expression levels of the ligand proteins used co-culture studies. Levels of one representative clone for each ligand protein were determined by cell surface biotinylation and quantitative analysis of Western blots after immunoprecipitation. Values for DLL1 and DLL4 see *Figure 4—source data 3*. ND: due to closely co-migrating background band protein levels could not be quantified. Surface expression validated by biotinylation of ES cells and antibody staining of PSMs.
DOI: https://doi.org/10.7554/eLife.40045.024

• Supplementary file 2. Relative ligand protein expression level in ES cell clones. The protein level of three independent clones used for co-culture studies was determined by quantitative analysis of Western blots and normalized to DLL1 clone #1 protein level measured in the same assay. Values for DLL1 and DLL4 see *Figure 4—source data 2*. ND: due to closely co-migrating background band protein levels could not be quantified.
DOI: https://doi.org/10.7554/eLife.40045.025

• Transparent reporting form
DOI: https://doi.org/10.7554/eLife.40045.026

### Data availability

All data generated or analysed during this study are included in the manuscript and supporting files and source data files.

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
