## [Decision Letter]

Thank you for submitting your article "The ectodomains determine ligand function in vivo and selectivity of DLL1 and DLL4 toward NOTCH1 and NOTCH2 in vitro" for consideration by *eLife*. Your article has been reviewed by three peer reviewers, and the evaluation has been overseen by Urban Lendahl as Reviewing Editor and Didier Stainier as Senior Editor. The following individuals involved in review of your submission have agreed to reveal their identity: Thomas Gridley (Reviewer #1); Rhett A Kovall (Reviewer #2); Cecilia Sahlgren (Reviewer #3).

The reviewers have discussed the reviews with one another and the Reviewing Editor has drafted this decision to help you prepare a revised submission.

Summary:

The manuscript addresses the important question of ligand-receptor specificity in the Notch signaling pathway. The authors describe structure-function analyses, and functional divergence, of the Notch ligands DLL1 and DLL4, using in vivo as well as in vitro techniques. The authors conclude that the DLL1 and DLL4 proteins activate the Notch1 and Notch2 receptors differently. DLL4 consistently activated NOTCH1 significantly better than DLL1, and DLL4 stimulated Notch2 significantly less efficiently than DLL1.To identify the domains of DLL1 and DLL4 that contributed to the observed differences in activating Notch1 and Notch2, domain swaps to generate a set of chimeric DLL1/DLL4 ligands was performed. These experiments revealed that the differences in activation potential of DLL4 and DLL1 toward Notch1 and Notch2 are encoded in the N-terminal part of the protein, comprising the MNNL-EGF3 domains, but also with a contribution from the third EGF repeat. This is interesting, as these regions lie outside the established receptor-ligand interaction domains.

Essential revisions:

1) In the Abstract the authors state, "Collectively, our data show that DLL4 and DLL1 preferentially activate NOTCH1 and NOTCH2, respectively,"; however, the N1/N2 ratio data in Figure 5A suggest that DLL1 has no preference for Notch1 over Notch2. Please clarify.

2) In the last paragraph of the subsection “Regions outside the known receptor binding domain are essential for full DLL1 function in vivo”, the authors use the word "undistinguishable", do the authors mean indistinguishable?

3) Regarding the in vivo data where different chimeras are used to substitute for DLL1 there appears to be additional staining or even puncta for the ligand, e.g. Figure 2Cd, Figure 3Cdfgi, Figure 6Bfh, that is not at the cell surface and is not seen in the wild-type tissues. Could the authors provide some explanation as to what this additional staining is and do they think that this accumulated ligand has any effect on the phenotypes that they are seeing?

4) In Figure 3Fc do the authors mean Dll1Dll4ki/Dll4ki rather than Dll1Dll1ki/Dll1ki as currently stated?

5) In Figure 5B, I would suggest that the authors label the domains of Notch1, Dll4, and Dll1 on the structures to clarify these regions for the reader who is not a structural biologist.

6) Figure 4E and F. Showing both black dots and triangles in the same graph is confusing.

7) Figure 5 and subsection “The region encompassing the MNNL up to and including EGF3 encodes the differential receptor selectivity of DLL1 and DLL4” MNNL-DSL (D1N-D_D4 & D4N-D_D1) values don't look intermediate but as low (if not lower) than D1N-E3_D4 – rephrase?

8) Discussion, second paragraph – D1ECD_D4ECD changed to D1ECD_D4ICD.

9) Much of the data (protein-protein levels, membrane levels and signalling activity) from the co-culture signalling studies are adjusted to a data point set to 1 and many of the analyses are further related to this reference value which is an average. There is high variability and the original data was not shown and the number of analyses were very high. I would like to see the original data, ask for a comparison between membrane levels, total protein levels and signal activity within a set of experiments to avoid making erroneous conclusions based on statistical "engineering"

10) Figure 4C, E and F Why only n=1 in Dll1? Does that represent the average? What is the variability?

11) Figure 5. Showing the N1 and N2 stimulations and not only the ratio could add some clarity (Despite varying membrane levels). Perhaps a supplementary figure?

12) Figure 5. How come D1N-E3 gives such a strong signal in reporter assays if it is not detectable/quantifiable on WB or biotinylation assay? Could you provide the image of the WB for assurance?

13) The chimeric ligand is expressed more efficiently than the endogenous one (Figure 3C). Could this affect the result?

The full reviews are also included for your reference, as they contain detailed and useful suggestions.

*Reviewer #1:*

The authors describe structure-function analyses, and functional divergence, of the Notch ligands DLL1 and DLL4, using cell co-culture assays, biochemical assays, and in vivo analyses of genetically modified mice expressing recombinant DLL1/DLL4 chimeric proteins. The authors demonstrate that, in cell based co-culture assays, the DLL1 and DLL4 proteins activate the NOTCH1 and NOTCH2 receptors differently. They generated mouse NOTCH1 and NOTCH2 reporter ES cell lines by integrating a Notch luciferase reporter in the Hprt locus of mouse E14TG2a ES (E14) cells, thus generating stable mN1 (N1rep) or mN2 (N2rep) reporter lines. These reporter lines were then co-cultured with ES cells with single copy integrations into the Hprt locus of the DLL1 or DLL4 coding sequences. Performing careful controls for total protein and cell surface protein expression levels, they showed that all DLL4 clones consistently activated N1 significantly better than all DLL1 clones, and all DLL4 clones stimulated mN2 significantly less efficiently than DLL1. To identify the domains of DLL1 and DLL4 that contributed to the observed differences in activating NOTCH1 and NOTCH2, they performed a series of domain swaps to generate a set of chimeric DLL1/DLL4 ligands. These experiments revealed that the differences in activation potential of DLL4 and DLL1 toward N1 and N2 are encoded in the N-terminal part of the protein, comprising the MNNL-EGF3 domains. Additional experiments showed that the third EGF repeat made an important contribution to receptor selectivity.

The authors then analyzed the importance of the amino acids that contact N1 in the binding interfaces of the MNNL and DSL domains in NOTCH receptor selectivity. In a series of biochemical experiments, the authors showed that swapping the contact residues of DLL4 onto DLL1 did not substantially affect Notch receptor activation compared to wildtype DLL1. Replacement of the DLL4 contact residues by the analogous residues of DLL1 slightly affected Notch receptor activation, but did not approach the activity difference of wildtype DLL1.

For in vivo analyses, the authors utilized a sophisticated genetic model system in which transgenes encoding chimeric DLL1/DLL4 ligands were introduced into the Hprt locus of E14 ES cells in a manner that requires Cre recombinase expression to permit expression of the chimeric ligand. Simultaneously, endogenous wildtype DLL1 expression was removed using a floxed Dll1 allele and a Cre driver (T(s):Cre) expressed in the primitive streak driven by a promoter derived from brachyury (T) locus. These experiments demonstrated that the extracellular domains of the DLL1 and DLL4 ligands regulated ligand function during somite formation. The authors then tested a chimeric ligand that contained the N-terminal region up to and including EGF3 of DLL1 fused to EGF4 and the remaining C-terminal portion of DLL4. Mice homozygous for this allele (and lacking the wildtype Dll1 allele) were still born and exhibited substantial axial skeletal defects, indicating that this chimeric allele cannot fully replace wildtype DLL1 function during development.

These experiments, as is characteristic of all the work from the Gossler and Blacklow laboratories, are carefully controlled, extremely thorough, and very convincing. This paper makes an important contribution to the literature on structure/function analysis of the Notch ligands.

*Reviewer #2:*

The authors submit an interesting and compelling manuscript that focuses on using different in vivo and cellular assays to characterize the functional differences between Dll1 and Dll4. The work is of high quality and the manuscript is well written. Moreover, their findings address an important area of Notch signaling and biology in general, and impact other recent publications that characterize functional differences between Dll1/4. If the authors were to address my relatively minor comments summarized below, I would recommend their manuscript to be published *eLife*.

In the Abstract the authors state, "Collectively, our data show that DLL4 and DLL1 preferentially activate NOTCH1 and NOTCH2, respectively,"; however, the N1/N2 ratio data in Figure 5A suggest that DLL1 has no preference for Notch1 over Notch2. Please clarify.

In the last paragraph of the subsection “Regions outside the known receptor binding domain are essential for full DLL1 function in vivo”, the authors use the word "undistinguishable", do the authors mean indistinguishable?

Regarding the in vivo data where different chimeras are used to substitute for DLL1 there appears to be additional staining or even puncta for the ligand, e.g. Figure 2Cd, Figure 3Cdfgi, Figure 6Bfh, that is not at the cell surface and is not seen in the wild-type tissues. Could the authors provide some explanation as to what this additional staining is and do they think that this accumulated ligand has any effect on the phenotypes that they are seeing?

In Figure 3Fc do the authors mean Dll1Dll4ki/Dll4ki rather than Dll1Dll1ki/Dll1ki as currently stated?

In Figure 5B, I would suggest that the authors label the domains of Notch1, Dll4, and Dll1 on the structures to clarify these regions for the reader who is not a structural biologist.

*Reviewer #3:*

The work by Tveriakhina and colleagues addresses the biochemical rational behind the context dependent functional differences of Notch ligands. The in vivo evidence is strong and they nicely demonstrate context dependent contributions of Notch ligand extracellular EGF repeats in vivo. They further demonstrate, although the data raise some concerns, that Dll1 and Dll4 ligands exhibit different receptor selectivity and discriminate between Notch1 and Notch2 receptors and that this is caused by differences in ligand ECD domains outside the established receptor-ligand interaction domains (contact domains). The paper is well written, clear and concise. The data is well presented and for the most part support the conclusions. The work provides novel insight into the receptor-ligand interphase and presents protein regions that contribute to functional divergence of ligands and warrants publication, provided the authors address the concerns raised.

---

## [Author Response]

Essential revisions:1) In the Abstract the authors state, "Collectively, our data show that DLL4 and DLL1 preferentially activate NOTCH1 and NOTCH2, respectively,"; however, the N1/N2 ratio data in Figure 5A suggest that DLL1 has no preference for Notch1 over Notch2. Please clarify.

Indeed, as the reviewer points out, DLL1 stimulates N1 and N2 (more-or-less) equally, whereas DLL4 preferentially activates NOTCH1 over NOTCH2. We have clarified this as follows:

“Collectively, our data show that DLL4 preferentially activates NOTCH1 over NOTCH2, whereas DLL1 is equally effective in activating NOTCH1 and NOTCH2, establish that the ectodomains dictate selective ligand function in vivo, and that features outside the known binding interface contribute to their differences.”

In addition, to stay within the length limit we have slightly reworded the Abstract as indicated.

2) In the last paragraph of the subsection “Regions outside the known receptor binding domain are essential for full DLL1 function in vivo”, the authors use the word "undistinguishable", do the authors mean indistinguishable?

We changed “undistinguishable” to “indistinguishable”.

3) Regarding the in vivo data where different chimeras are used to substitute for DLL1 there appears to be additional staining or even puncta for the ligand, e.g. Figure 2Cd, Figure 3Cdfgi, Figure 6Bfh, that is not at the cell surface and is not seen in the wild-type tissues. Could the authors provide some explanation as to what this additional staining is and do they think that this accumulated ligand has any effect on the phenotypes that they are seeing?

The cytoplasmic staining of ligands including wt DLL1 (Figure 3Cjl and 6Bdh) reflects at least in part the presence of ligands in the ER and Golgi apparatus as has been shown previously by co-staining in cell lines (Geffers et al., 2007 and Müller et al., 2014) and has previously also been observed in the PSM (Preusse et al., 2015). We have added this information in the legend to Figure 2C as follows:

“Additional intracellular staining most likely reflects the presence of the ligand in the ER and trans Golgi as observed previously for DLL1 in cultured cells (Geffers et al., 2007; Müller et al., 2014) and for endogenous DLL1 and transgenic DLL4 in the PSM (Preusse et al., 2015)“.

And the new citation (Müller et al., 2014) was added to the references.

4) In Figure 3Fc do the authors mean Dll1Dll4ki/Dll4ki rather than Dll1Dll1ki/Dll1ki as currently stated?

We regret this error and have corrected the mislabeling.

5) In Figure 5B, I would suggest that the authors label the domains of Notch1, Dll4, and Dll1 on the structures to clarify these regions for the reader who is not a structural biologist.

We have labelled the structures in revised Figure 5B as suggested and indicated the domains by different shadings.

6) Figure 4E and F. Showing both black dots and triangles in the same graph is confusing.

We have changed Figure 4E and F (please see also point 9).

7) Figure 5 and subsection “The region encompassing the MNNL up to and including EGF3 encodes the differential receptor selectivity of DLL1 and DLL4” MNNL-DSL (D1N-D_D4 & D4N-D_D1) values don't look intermediate but as low (if not lower) than D1N-E3_D4 – rephrase?

We have rephrased the sentence to clarify that the chimeric pairs exhibit equivalent N1/N2 ratios as follows:

“When chimeras include the MNNL-EGF2 or MNNL-DSL region of one ligand and the remainder of the other, the N1/N2 stimulation ratios of the chimeric pairs are equivalent (Figure 5A), indicating that the third EGF-like repeat makes an important contribution to receptor selectivity.”

8) Discussion, second paragraph – D1ECD_D4ECD changed to D1ECD_D4ICD.

This error has been corrected.

9) Much of the data (protein-protein levels, membrane levels and signalling activity) from the co-culture signalling studies are adjusted to a data point set to 1 and many of the analyses are further related to this reference value which is an average. There is high variability and the original data was not shown and the number of analyses were very high. I would like to see the original data, ask for a comparison between membrane levels, total protein levels and signal activity within a set of experiments to avoid making erroneous conclusions based on statistical "engineering"

We have included the ratios of DLL1/TUB and DLL4/TUB determined by Western blot quantitation in Figure 4—source data 2 and show now an additional graph (in the revised Figure 4—figure supplement 1) depicting the values obtained from multiple technical replicates of the DLL1 and DLL4 clone done in parallel.

We also show in the revised Figure 4E and F the non-normalized N1 and N2 activation (RLU values minus background; previously only given in the tables in Figure 4—figure supplement 1—source data 2; now for clarity also in Figure 4—source data 4 and 5) from analyses done in parallel in addition to the normalized values. This representation shows that despite the variability between different co-cultures DLL4 consistently activates N1 better and N2 less efficiently than DLL1.

In addition we have expanded Figure 4 figure supplement 1 and show graphs for the technical replicates of all DLL1 (4) and DLL4 (10) clones.

Normalized values are presented in the right graphs of Figure 4E and F, and in the right graphs in Figure 4—figure supplement 1.

10) Figure 4C, E and F Why only n=1 in Dll1? Does that represent the average? What is the variability?

The dot in Figure 4 C represented the “average” of 10 DLL1 measurements set to 1. Each DLL4 value was referenced to its paired DLL1 value, which was arbitrarily set to 1 for each measurement. As described for point 9 we have included now the original DLL/TUB ratios in Figure 4—source data 2 and Figure 4—figure supplement 1A.

11) Figure 5. Showing the N1 and N2 stimulations and not only the ratio could add some clarity (Despite varying membrane levels). Perhaps a supplementary figure?

We followed this suggestion and have included new Figure 5—figure supplement 1 showing the relative luciferase activity measured in co-cultures.

12) Figure 5. How come D1N-E3 gives such a strong signal in reporter assays if it is not detectable/quantifiable on WB or biotinylation assay? Could you provide the image of the WB for assurance?

D1N-E3 is detectable on WB and after biotinylation, but cannot be quantified with confidence due to a closely co-migrating background band detected by the anti-Flag antibodies. We have tried different polyacrylamide concentrations but could not separate the background band enough for reliable quantification. We include representative WBs (which could be added as a supplementary file if required) as Author response image 1.

**Author response image 1. respfig1:** Expression of D1*N-E3_*D4 cannot be quantified with confidence due to a closely co-migrating background band detected by the anti-Flag antibody. (**A**) Expression of wild type DLL1 and DLL4 and of chimeric ligand proteins. Protein migration size varies due to differential post-translational modifications of the extracellular domains of DLL1 and DLL4. Double bands in DLL4, D4*ECD*_D1/*CD*, D4*N-E3_*D1, D4*N-E2*D1, and D4*N-D_*D1 lysates represent most likely differential modification states of the DLL4’s ECD. (**B**) D1*N-E3_*D4 migrates right at the dye front when using a 5% polyacrylamide gel. (**C**) Immunoprecipitation of D1*N-E3_*D4 after cell surface biotinylation in four independent assays. Input samples represents the whole cell lysate, IP samples represents the immunoprecipitated proteins. D1*N-E3_*D4 is detected in IP samples indicating its presence at the cell surface. The expression, however, cannot be quantified with confidence as the proteins are detected right at the dye front when using a 5% polyacrylamide gel. Pink arrowheads point to the proteins of interest. Green rectangles highlight the D1*N-E3_*D4 chimera lysate in (**A**) and (**B**).

13) The chimeric ligand is expressed more efficiently than the endogenous one (Figure 3C). Could this affect the result?

We appreciate the reviewer’s concern about whether there are different amounts of protein in the two conditions. However, because anti-DLL1 antibodies were used to detect endogenous DLL1, and anti-DLL4 antibodies to detect the chimeric ligand (as indicated in the panels of Figure 3C), which cannot be detected using the anti-DLL1 antibodies, the protein levels cannot be compared based on differences in staining intensity because the antibodies are different.

In addition, we have corrected spelling/formatting mistakes and errors, updated legends and references to figures, figure supplements and source data files, included the Key Resources Table and a statement concerning mycoplasma tests and cell line authentication in the Materials and methods section, and made other changes requested by the editorial support team (supplementary files changed to figure supplements, file type changes, adjustment of nomenclature, added information in the Materials and methods section).